# FEW-SHOT SPECIES RANGE ESTIMATION

## ABSTRACT

Understanding where a particular species can or cannot be found is crucial for ecological research and conservation efforts. By mapping the spatial ranges of all species on Earth, we could obtain deeper insights into how global biodiversity is affected by climate change and habitat loss. However, accurate range estimates are available for a relatively small proportion of known species. For most species, we have only have a few prior observations indicating the locations where they have been previously recorded. In this work we address the challenge of training with limited observations by developing a new approach for few-shot species range estimation. During inference, our model takes a set of spatial coordinates as input, along with optional metadata such as text, and outputs a species encoding that can be used to predict the range of a previously unseen species in feed-forward manner. We validate our method on two challenging benchmarks, where we obtain state-of-the-art performance in predicting the ranges of unseen species, in a fraction of the compute time, compared to recent alternative approaches.

## 1 INTRODUCTION

Understanding the spatial distribution of plant and animal species is essential to mitigate the ongoing decline in global biodiversity (Jetz et al., 2019). Monitoring these distributions over time allows us to quantify the effects of climate change, habitat loss, and conservation interventions (Mantyka-pringle et al., 2012). An estimate of the species' distribution typically starts with a collection of location data where the species is confirmed to be present. Traditionally, this data is used to train a models that generate detailed predictions over a region of interest (Elith et al., 2006; Beery et al., 2021). When sufficient data is available, these models enable practitioners to estimate important quantities such as the spatial range (i.e., where a species can be found) or abundance (i.e., the total number of individuals) of a species, in addition to quantifying how these quantities are changing over time.

Despite the availability of well-established modeling techniques, our current understanding of species' distributions is extremely limited due to little or no observational data being available for most species. For example, iNaturalist, one of the largest citizen science platform documenting global biodiversity, has collected over 130 million "research grade" observations for approximately 373,000 species globally (iNaturalist, 2024). However, the data is severely long-tailed: a small percentage of common species account for the majority of the observations, while many species have very few observations. In fact, over half of the 373,000 species catalogued by iNaturalist have been observed fewer than 10 times. This data limitation is amplified by the fact that there are estimated to be several million species on earth, many of which are not yet documented by science (Mora et al., 2011). Identifying locations where under-observed species can be found is a time consuming and laborious process, often requiring long expeditions in remote locations searching for species that are hard to find. Consequently, there is a pressing need for computational methods that can reliably estimate the spatial distributions of species using only a small number of observations.

Knowing the range of one species can help predict the range of another due to shared ecological, environmental, and geographic contexts. Recent advances in range estimation have leveraged this idea by training shared models using millions of observations across tens of thousands of species (Cole et al., 2023). However, these models still rely on relatively large numbers of training observations for individual species, which limits their applicability to species with sparse observations. In this work, we introduce a novel Transformer-based model that overcomes this limitation and offers two key advantages over previous approaches. First, our method achieves superior performance in the

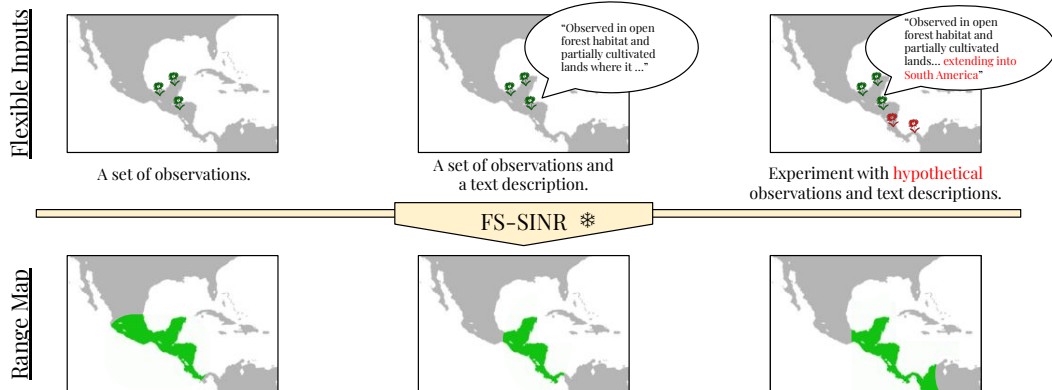

Figure 1: **Few-shot species range estimation**. We introduce, FS-SINR, a new approach for few-shot species range estimation. FS-SINR is trained on citizen science species collected location data, and once trained, it can be used to estimate the spatial range of an unseen species with a single forward pass through the model, i.e., no retraining is required at inference time. Furthermore, it supports different input modalities such as variable length sequences of geographic locations in addition to other metadata such as text.

few-shot regime, a scenario that represents the reality for the majority of species but has been underexplored in prior research. Second, our model can make accurate predictions for species not present in the training set without any additional training, which can enable interactive exploration and modeling. At inference time, we only require a set of observed locations for the unseen species to generate reliable range estimates. Furthermore, we show that our model can flexibly incorporate additional non-geographic context information (e.g., a text summary of the species' habitat or range preferences) to further improve prediction quality. Fig. 1 provides an overview of how our method can be used at inference time.

In summary, we make the following core contributions: (i) We introduce FS-SINR, a new approach for few-shot species range estimation. FS-SINR has novel capabilities, including the ability to predict the spatial range of a previously unseen species at inference time without requiring any retraining. (ii) We demonstrate, across two challenging benchmark datasets, that FS-SINR achieves state-of-the-art performance in the few-shot setting.

## 2 RELATED WORK

**Species Distribution Modeling**. Estimating the spatial distribution of species has been a widely explored topic both in statistical ecology and machine learning (Beery et al., 2021). The goal is to develop models that can predict the distribution of species in space, and possibly time, given sparse observation data. In the context of machine learning, different approaches based on traditional techniques such as decision trees have been extensively explored (Phillips et al., 2004; Elith et al., 2006). More recently, several deep learning methods have been introduced for the task (Botella et al., 2018; Mac Aodha et al., 2019; Kellenberger et al., 2024). One of the strengths of deep methods is that they can jointly represent thousands of different species inside of the same model and have been shown to improve as more training data is added. For example, in SINR (Cole et al., 2023), the authors demonstrated that range estimation performance improves as more data from different species is added.

There has also been work investigating different approaches for addressing some of the challenges associated with training and evaluating these models. Examples include attempts to addresses imbalances across species in the training observation data (Zbinden et al., 2024b), methods for sampling pseudo-absence data (Zbinden et al., 2024a), biases in the training locations (Chen & Gomes, 2019), representing location information (Rußwurm et al., 2024), discretizing continuous model predictions (Dorm et al., 2024), using additional metadata such as images (Teng et al., 2023; Dollinger et al., 2024; Picek et al., 2024), and designing new evaluation datasets to benchmark

performance (Cole et al., 2023; Picek et al., 2024). In our work, we investigate the under-explored few-shot setting, where only limited observations (e.g., fewer than ten) are available for each species.

**Few-shot Species Range Estimation**. There are several aspects of the species range estimation task in the low data regime that makes it different from other few-shot applications more commonly explored in the literature (Parnami & Lee, 2022; Wang et al., 2020). For one, the input domain is fixed (i.e., all the locations on earth), each location can support more than one species (i.e., multi-label as opposed to multi-class), the label space is much larger (i.e., tens of thousands of species as opposed to hundreds of classes in image classification), and only partial supervision is available (i.e., we only have presence data, with no confirmed absences).

Some attempts have been make at training species range estimation models using limited amounts of observation data. Cole et al. (2023) demonstrated that their SINR approach performs much worse when trained on at most ten observations per species compared to training larger amounts. Lange et al. (2023) proposed an active learning-based approach for estimating the ranges of previously unseen species. They performed experiments in the low data regime, but in contrast to us, they require confirmed absence observations, in addition to presences, when updating their model for an unseen species. The zero-shot setting, i.e., where no location observations have been observed, has also been explored. Specifically, LD-SDM (Sastry et al., 2023) used text information to encode the taxonomic knowledge and LE-SINR (Hamilton et al., 2024) used text describing a species' range or preferred habitat. At inference time, these *zero-shot* methods can make predictions for previously unseen species even when no observation (i.e., location) information was available, but when text is. LE-SINR performed few-shot experiments whereby they used a language encoder to estimate an initial encoding for a species and combined it with a linear classifier that needs to be trained to generate range predictions. In contrast, our FS-SINR approach does not require any additional training to make predictions for previously unseen species at inference time. We compare to LE-SINR in our evaluation and demonstrate that we outperform it in both the zero and few-shot settings and also show that free-form text is superior to the taxonomic text used in LD-SDM.

## 3 METHODS

In this section we first set up the species range estimation problem and then describe our approach for few-shot range estimation.

### 3.1 SPECIES RANGE ESTIMATION

We start by describing the SINR approach from Cole et al. (2023). Let $\boldsymbol{x} = (\text{lat}, \text{lon}) \in \mathcal{X}$ be a location of interest sampled from a spatial domain $\mathcal{X}$ (e.g., the surface of the earth). Our goal is train a model $g() : \mathcal{X} \to [0, 1]^s$ to predict the probability of $s$ different species of interest occurring at $\boldsymbol{x}$. We will write $\hat{\boldsymbol{y}} = g(\boldsymbol{x})$, where $\hat{y}_j \in [0, 1]$ (the $j^{th}$ entry of $\hat{\boldsymbol{y}}$) denotes the probability that species $j$ occurs at location $\boldsymbol{x}$.

We decompose the model as $g() = h_{\boldsymbol{\phi}}() \circ f_{\boldsymbol{\theta}}()$, where $f_{\boldsymbol{\theta}}() : \mathcal{X} \to \mathbb{R}^d$ is a location encoder with parameters $\boldsymbol{\theta}$ and $h_{\boldsymbol{\phi}}() : \mathbb{R}^d \to [0, 1]^s$ is a multi-label classifier with parameters $\boldsymbol{\phi}$. The location encoder $f_{\boldsymbol{\theta}}()$ maps a location $\boldsymbol{x}$ to a $d$-dimensional latent embedding $f_{\boldsymbol{\theta}}(\mathbf{x})$. The multi-label classifier $h()$ is implemented as a per-species linear projection followed by an element-wise sigmoid non-linearity, meaning that $\hat{\boldsymbol{y}} = \sigma(f_{\boldsymbol{\theta}}(\boldsymbol{x})\boldsymbol{W})$, where $\boldsymbol{W} \in \mathbb{R}^{d \times s}$ (i.e., $h_{\boldsymbol{\phi}}() = \boldsymbol{\phi} = \boldsymbol{W}$) and $\sigma()$ is the sigmoid function. Thus, each column vector $\boldsymbol{w}_s$ of $\boldsymbol{W}$ can be viewed as a species embedding, which we can combine with a location embedding $f_{\boldsymbol{\theta}}(\boldsymbol{x})$ in an inner product to compute the probability that the species $s$ is present at $\boldsymbol{x}$. Importantly, the location embedding is shared across all species. Once trained, it is then possible to generate a prediction for a given species for all locations of interest (e.g., the entire surface of the earth) by evaluating the model at all locations (i.e., $\boldsymbol{x} \in \mathcal{X}$).

One of the main challenges associated with training models for species range estimation is that there is a dramatic asymmetry in the available training data. Specifically, it is much easier to collect presence observations (i.e., confirmed sightings of a species) than absence observations (i.e., confirmation that a species is not present at a specific location). As a result, many methods have been developed to train models using *presence-only* data. In the presence-only setting, we have access to training pairs $(\boldsymbol{x}, z)$, where x is a geographic location and $z \in \{1, \dots, s\}$ is an integer

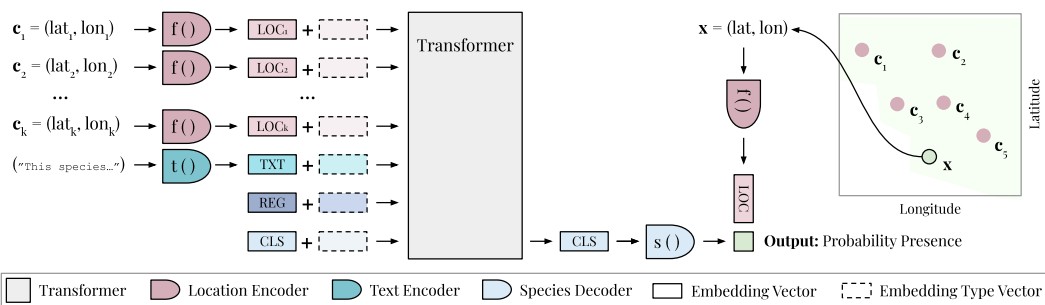

Figure 2: **FS-SINR overview**. Here we depict our few-shot species range estimation model. The input consists of an arbitrary number of context locations $\mathcal{C}^s$, that are each independently tokenized using a location encoder $f_{\boldsymbol{\theta}}()$, and optional auxiliary context information like text. A register token (REG) Darcet et al. (2024) and a class token (CLS) are appended to the input as well. All input tokens are processed by a Transformer $m_{\boldsymbol{\psi}}()$. To make a prediction at a query location $\boldsymbol{x}$, we compute the embedding of $\mathbf{x}$ using the location encoder and the projected embedding of the CLS token which is output from the species decoder MLP $s()$.

indicating which species was observed there. To overcome the lack of confirmed absence data, one common approach is to generate *pseudo-absences* by sampling random locations on the surface of the earth (Phillips et al., 2009). Give these pseudo-absences, the parameters of $g()$ can be trained in an end-to-end manner using variants of the cross entropy loss. In this, work we use the *full assume negative loss* (i.e., $\mathcal{L}_{\mathrm{AN-full}}$) from Cole et al. (2023) to train the SINR baseline:

$$\mathcal{L}_{\mathrm{AN\text{-}full}}(\hat{\mathbf{y}}, z) = -\frac{1}{S} \sum_{j=1}^{S} \left[ \mathbb{1}_{[z=j]} \lambda \log(\hat{y}_j) + \mathbb{1}_{[z \neq j]} \log(1 - \hat{y}_j) + \log(1 - \hat{y}'_j) \right], \quad (1)$$

where $z$ is the index of the species present for a given training instance, $\hat{y}_j$ is the predicted probability of the presence of species $j$, $\hat{y}'_j$ is the model prediction for a randomly sampled pseudo-absence location, and $\lambda$ is a hyperparameter that balances the presence and pseudo-absence loss components.

## 3.2 FEW-SHOT RANGE ESTIMATION

For the SINR model to make predictions for a new species, it is necessary to learn a new embedding vector $\boldsymbol{w}_s$ for that species. If additional location data is later observed for that species, the model must be updated again. However, the number of observations, with associated locations, for uncommon species can be limited and thus it is necessary to have methods that can be updated efficiently with limited training data.

We address this challenge by proposing a new approach for few-shot species range estimation called FS-SINR. Our model can predict the probability of presence for a previously unobserved species directly at inference time given only the set of confirmed presence locations available, without any retraining or parameter updates. At inference time we assume we have access to a set of context locations $\mathcal{C}^s = \{\boldsymbol{c}_1, \ldots, \boldsymbol{c}_k\}$, which represent a set of $k$ locations where the species $s$ has been confirmed to be present. Each entry is this set represents a geographic location, i.e., $\boldsymbol{c} = (\mathrm{lat}, \mathrm{lon})$. Like SINR, our model is also conditioned on a location $\boldsymbol{x}$ of interest (i.e., the 'query' location), but uses the context locations to inform the prediction for the query location. Note, these locations can come from a species that was not previously seen by the model during training.

We represent our FS-SINR model as $g(\boldsymbol{x}) = m_{\boldsymbol{\psi}}(f_{\boldsymbol{\theta}}(\boldsymbol{x}), \mathcal{C}^s)$. Unlike in SINR where the classifier head $h_{\boldsymbol{\phi}}()$ is a simple multi-label classifier and sigmoid non-linearity, in our case the 'head' of the model $m_{\boldsymbol{\psi}}()$ is a Transformer-based encoder (Vaswani et al., 2017). An illustration of FS-SINR is depicted in Fig. 2. The model takes an unordered set of context locations $\mathcal{C}^s$ as input, where each location is encoded into an embedding vector (i.e., token) via a SINR-style multi-layer perceptron location encoder. Importantly, our model is invariant to the number and ordering of the context locations as we do not append any positional embeddings. This flexibility ensures that our model

can process a variable number of context locations at inference. We also append an additional register token (REG) as in (Darcet et al., 2024) to provide the model with an additional token to 'store' useful information. Given that the input sequence is unordered and may or may not include additional context information, we also add additional learned 'embedding type' vectors to each token such that the Transformer knows if a given input token is a location, or register, text, etc.

We represent the species embedding vector (i.e., $\boldsymbol{w}_s$ in SINR) as the class token CLS of the Transformer after passing it through a small species decoder MLP $s()$. To make a final prediction, we simply compute the inner product between the location embedding of the query location $\boldsymbol{x}$ and the species embedding vector, and pass it through a sigmoid. Our approach is computationally efficient in that once the species embedding is generated once it can then be efficiently multiplied by the embeddings for all locations of interest to generate a prediction for a species' range.

FS-SINR uses a similar training loss to $\mathcal{L}_{\mathrm{AN-full}}$. However as FS-SINR has no equivalent to $h_\phi()$ we cannot easily include all species in the loss and instead consider only those within the same batch of training examples $S^b$. These species will have generated a species embedding vector during the forward pass which can be used to predict probabilities of presence for that species for all locations in the batch. We denote this modification as $\mathcal{L}_{\mathrm{AN-full-b}}$, which indicates that we are considering only those elements contained within the current batch $b$:

$$\mathcal{L}_{\mathrm{AN\text{-}full\text{-}b}}(\hat{\mathbf{y}}, z^b) = -\frac{1}{S^b} \sum_{j=1}^{S^b} \left[ \mathbb{1}_{[z^b=j]} \lambda \log(\hat{y}_j) + \mathbb{1}_{[z^b \neq j]} \log(1 - \hat{y}_j) + \log(1 - \hat{y}'_j) \right]. \quad (2)$$

### 3.2.1 ADDITIONAL CONTEXT INFORMATION

The design of FS-SINR is flexible, in that we can also provide additional context information to the model if it is available. For example, if there is additional text (e.g., a range description) or visual (e.g., images) information available for a novel species it could be added to the context, assuming such information was also available at training time for other species. This observation is inspired by recent work that also uses language derived information to improve range predictions (Sastry et al., 2023; Hamilton et al., 2024). This additional information can provide a rich source of metadata encoding aspects of a species' habitat preferences, even when there might only be a limited number of location observations available for it. We can represent the expanded context vector as $\mathcal{C}^s = \{\boldsymbol{t}_s, \boldsymbol{c}_1, \ldots, \boldsymbol{c}_k\}$, where $\boldsymbol{t}_s$ denotes a fixed length text embedding from a large language model extracted for species $s$. While, not explored in this work it is also possible to include addition context modalities such as a fixed-length embedding vector from a pretrained vision model.

## 4 EXPERIMENTS

Here we evaluate FS-SINR on species range estimation and compare it to existing methods.

### 4.1 IMPLEMENTATION DETAILS

Our location encoders use the same fully connected neural network with residual connections as in (Cole et al., 2023). Each of the context locations is processed by the same shared location encoder which is first pretrained as in SINR after which the multi-label classifier head is discarded. Importantly, this pretrained encoder is only trained on species from the training set, and does not observe any data from the evaluation species. The text embedding backbone is a frozen GritLM (Muennighoff et al., 2024) which provides a fixed length embedding vector. We train a small two layer fully connected text encoder to transform this into the text token. The Transformer contains four encoder layers and the parameters are updated jointly with the location and text encoders and species decoder during training. In total, our model has 6.3M learnable parameters compared to 11.9M for SINR. We train with a batch size of 2,048 instances and randomly drop-out text or location tokens during training with a probability of 0.2 and 0.1 respectively to enhance robustness.

We train our model using the presence-only dataset from (Cole et al., 2023) which contains 35.5 million citizen science observations for 47,375 species from the iNaturalist platform (iNaturalist, 2024). During training we supply our model with 20 context locations per training example, though

we find that the model performance is very robust to the number of context locations provided during training. We evaluate models using the IUCN and S&T datasets, which contain 2,418 and 535 expert and model-derived binary species range maps respectively. The IUCN dataset is more globally distributed and contains a larger variation in range size across species, while the S&T dataset only contains bird species that are primarily, but not always, found in North America and have a larger range size. We note that the evaluation datasets used could contain errors, but they represent the currently best available data and contain large variety in terms of range size and location. Importantly, unless otherwise stated, we hold out any species from the union of these two datasets from the training set so that species from the evaluation set are not observed during training. As a result, by default, our model is trained data from 44,422 species. Performance is reported in terms of mean average precision (MAP).

At inference time, generating a species' range for our FS-SINR model for a held-out species only requires a single forward pass through the model to get an embedding vector for the species. Current methods cannot be used in such a feed forward manner and need to be retrained for each species that were not observed at training time. To obtain an equivalent embedding for our baselines (i.e., SINR and LE-SINR) we train a per-species binary logistic regression classifier using any few-shot presence observations that are available in addition to adding 10,000 uniformly random and 10,000 target (i.e., in locations where species are) pseudo-absences as in Hamilton et al. (2024). For fairness, we keep the presence observations consistent across each method and the larger number of presences are supersets of the smaller ones. Additional implementation details are provided in Appendix A.

## 4.2 FEW-SHOT EVALUATION

First we evaluate how effective different range estimation models are at few-shot range estimation. The goal for each model is to generate a plausible prediction for a previously unseen species' range given limited location observations. Quantitative results are presented in Fig. 3.

The SINR (Cole et al., 2023) baseline performs poorly in the low data regime, but as more data is added performance improves. As noted earlier, here a per-species embedding vector is learned using logistic regression using the provided presence samples and generated pseudo-absences. The recently introduced LE-SINR (Hamilton et al., 2024) approach extends the basic SINR model to use text information, when available, at inference time. We can see that when any form of text data is available, LE-SINR outperforms SINR.

In all instances, when the same metadata is available, FS-SINR outperforms existing methods. Furthermore, we also outperform SINR in the larger data regime (i.e., when 50 observations are available). Importantly, unlike the baselines we compare to, FS-SINR does not need to be retrained at inference time. Instead, it can make predictions in a feed forward manner irrespective of the context data available. This is advantageous in interactive settings, whereby the model can compute the SINR locations encodings for all query locations on earth once and then the user could experiment by adding different context information interactively.

We present qualitative results for three different species in Fig. 4 where we visualize FS-SINR's predictions as we change the number of context locations. Given only a single context location, the model does a sensible job at localizing the species on the earth. This supports the findings from Fig. 3 where we observe strong performance even when only one context location is available. When more information is provided, the predicted range more closely resembles the expert-derived range shown in the first row. However, we do note that the model can still make mistakes in our low data setting, such as the erroneous predictions for the 'Black and White Warbler' in South America. In Fig. 5 we illustrate some examples of how text information, when paired with limited context locations, can influence the model predictions. Here we observe dramatically different predicted ranges when the text prompt encourages the model to focus on different habitat types. We note that each of the predicted ranges are still consistent with the location of the single context location provided. Additional qualitative examples are provided in the appendix.

## 4.3 ZERO-SHOT EVALUATION

In addition to being able to generate range predictions in the few-shot setting when limited location observations are provided, our approach is also able to make predictions when no location informa-

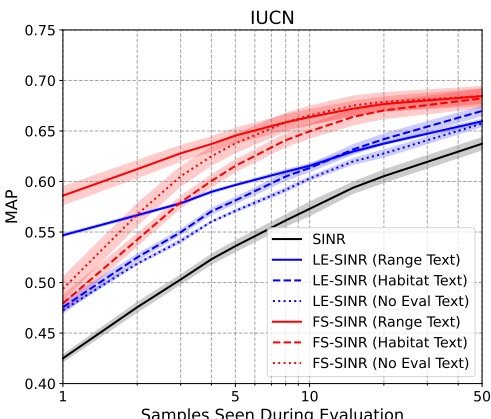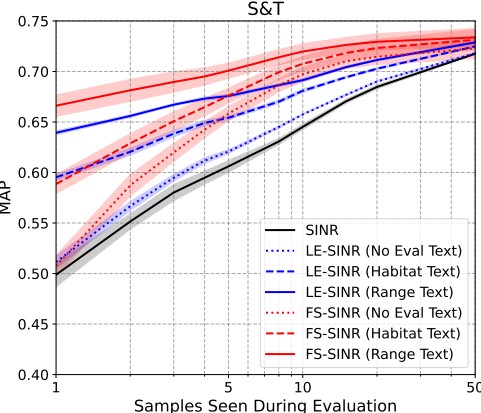

Figure 3: **Few-shot results**. Here we evaluate different models on the task of species range estimation on the IUCN (left) and S&T (right) datasets. On the x-axis we vary the amount of location observations (i.e., samples) seen at inference time for the held-out evaluation species. The y-axis represents MAP, where higher values are better. The error bars display the standard deviation of three different runs. Our FS-SINR approach outperforms existing methods. Note, except for FS-SINR, all other models need to be retrained during evaluation when more samples are provided. We include these results in tabular form in Tabs. A3 and A4 in the Appendix.

tion is provided, i.e., the *zero*-shot setting. These zero-shot results are presented in Tab. 1 for both the IUCN and S&T datasets.

We present results for several variants of FS-SINR where different types of text metadata data are used. As a baseline, we also present the performance of SINR (row 1) where the the evaluation species are part of its training set. We can also add data from these species to the training set of our approach which unsurprisingly boosts performance (e.g., row 3 vs. 9), though unlike SINR, FS-SINR does not have weights associated with individual species and so the impact of seeing evaluation species during training is fairly small. As a trivial baseline, we also report performance of FS-SINR (row 4) when no location or text metadata are provided, i.e., this is simply the output of the class token. As expected, this model performs poorly, but interestingly it seems to have learned some spatial prior that results in non-trivial predictions on S&T which contains bird species mostly concentrated in North America. We also compare to a version of FS-SINR (row 5) where we use taxonomic text as in LD-SDM (Sastry et al., 2023) (see Appendix B.7 for further details). In all instances our FS-SINR approach we outperform LE-SINR (Hamilton et al., 2024), even though both models are provided with the same information at inference time (e.g., row 8 vs. 9). Confirming observations in LE-SINR we see that range text is more informative than habitat text (e.g., row 7 vs. 9). Additional zero-shot results can be found in Tab. A1, where we evaluate different input features and location encoders.

### 4.3.1 ABLATIONS

We provide additional ablation experiments for FS-SINR in the appendix. We present results with different input features and location encoders. We also evaluate the impact of the amount of data used to train FS-SINR and pretrain the SINR location encoder we use. Finally, we also explore architectural modifications such as removing the final species decoder that operates on the output of the Transformer. We observe that FS-SINR is robust to many of these changes, justifying the design decision we made.

## 5 LIMITATIONS

While FS-SINR exhibits impressive zero and few-shot performance, there are several notable limitations. First, given a set of input context locations FS-SINR is deterministic in that it will always generate the same output range map. In practice, in the few-shot regime, the same set of points

Table 1: **Zero-shot results**. We compare to SINR (Cole et al., 2023) and LE-SINR (Hamilton et al., 2024) in the zero-shot setting where no location information is provided to each model. We denote additional metadata used by models as RT for 'Range Text' and HT for 'Habitat Text'. TST represents 'Test Species in Train', indicating that a model uses location observations for the evaluation species at training time, unlike other models where these species are excluded. 'TRT' indicates the model was trained using taxonomy rank text as in (Sastry et al., 2023), and is provided with the full taxonomic description from 'class' to 'species' during evaluation. SINR cannot make zero-shot predictions, thus the results presented for it is the performance on the evaluation set when these species have been observed at training time. This provides an upper bound on performance. Results are presented as MAP, where higher is better.

| ID | Method | Variant | IUCN | S&T |
|----|--------|---------|------|------|
| 1 | SINR | TST | 0.67 | 0.77 |
| 2 | FS-SINR | HT, TST | 0.38 | 0.59 |
| 3 | FS-SINR | RT, TST | 0.55 | 0.67 |
| 4 | FS-SINR | | 0.05 | 0.18 |
| 5 | FS-SINR | TRT | 0.21 | 0.34 |
| 6 | LE-SINR | HT | 0.28 | 0.52 |
| 7 | FS-SINR | HT | 0.33 | 0.53 |
| 8 | LE-SINR | RT | 0.48 | 0.60 |
| 9 | FS-SINR | RT | 0.52 | 0.64 |

could actually be representative of many different possible range maps. An obvious, and easy to implement, extension of our work is to introduce stochasticity into the model outputs, e.g., by treating class token output from the Transformer as a latent embedding for an additional sampling step. In Fig. A15 we obverse that initializing FS-SINR with different random seeds during training results in diverse range predictions across the different models. We leave this for future work. Second, at inference time, users may wish to provide example locations indicating where a specific species has *not* been found, i.e., confirmed absences. Currently our model is trained using presence-only data, but could be adapted to use absence information, if available, which could be denoted via a different embedding type vector which can be learned during training alongside our existing token type embeddings. However, obtaining large-scale reliable absence data for tens of thousands of species is a challenging task. Finally, global-scale citizen science datasets like the one we use to train FS-SINR can contain large biases (Geldmann et al., 2016), e.g., location, temporal, or taxonomic biases, among others. We do not explicitly account for these biases during training, and thus we would caution the use of the predictions of our model in any applications that would use our range predictions in the context of biodiversity assessments. However, we note that we outperform existing recent state-of-the-art range estimation methods, especially in the low observation data setting, and do not require any retraining at inference time.

## 6 CONCLUSION

We have limited knowledge regarding the geographic distributions of the majority of species on earth. This lack of understanding is further hampered by the fact that we also have insufficient data to train models to estimate their ranges. To address this problem, we introduced FS-SINR, a new approach for few-shot species range estimation. We demonstrated that our approach is naturally able to fuse data from different modalities and at inference time can make plausible predictions for the ranges of previously unseen species. Our quantitative analysis, using expert-derived range maps, shows that we obtain a 5% to 10% improvement in performance compared to current approaches in the low data setting for previously unseen species, e.g., when the number of observations equals ten. Additionally, we also outperform existing methods in the zero-shot setting. While our results are promising, they also indicate that there are many potential opportunities for future improvements in this important topic.

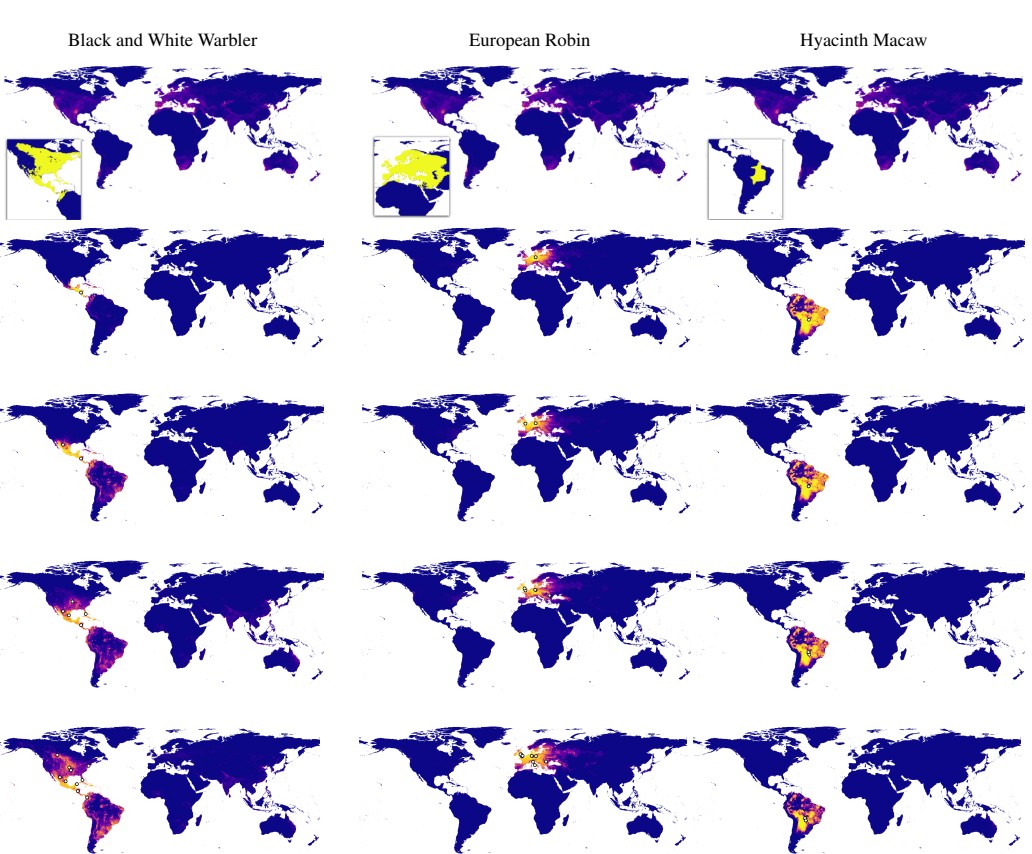

Figure 4: **Few-shot range estimation with increasing context locations**. Here we illustrate FS-SINR's few-shot range predictions given a set of context locations $\{0, 1, 2, 5, 10\}$ and no text descriptions for the `Black and White Warbler` (left), `European Robin` (center), and the `Hyacinth Macaw` (right), with expert-derived range maps inset. In the first row, we show the expert-derived range inset and the prediction for the model when no context locations are provided (which is the same for all species). Then, in the remaining rows we increase the number of context locations, denoted as 'o'. Zoom in to see the context locations. Most of the `Hyacinth Macaw` range is within the Amazon rainforest where we have few observations, and so most of our data comes from the same locations around human settlements. Providing many very similar observations does not seem to impact the predicted range.

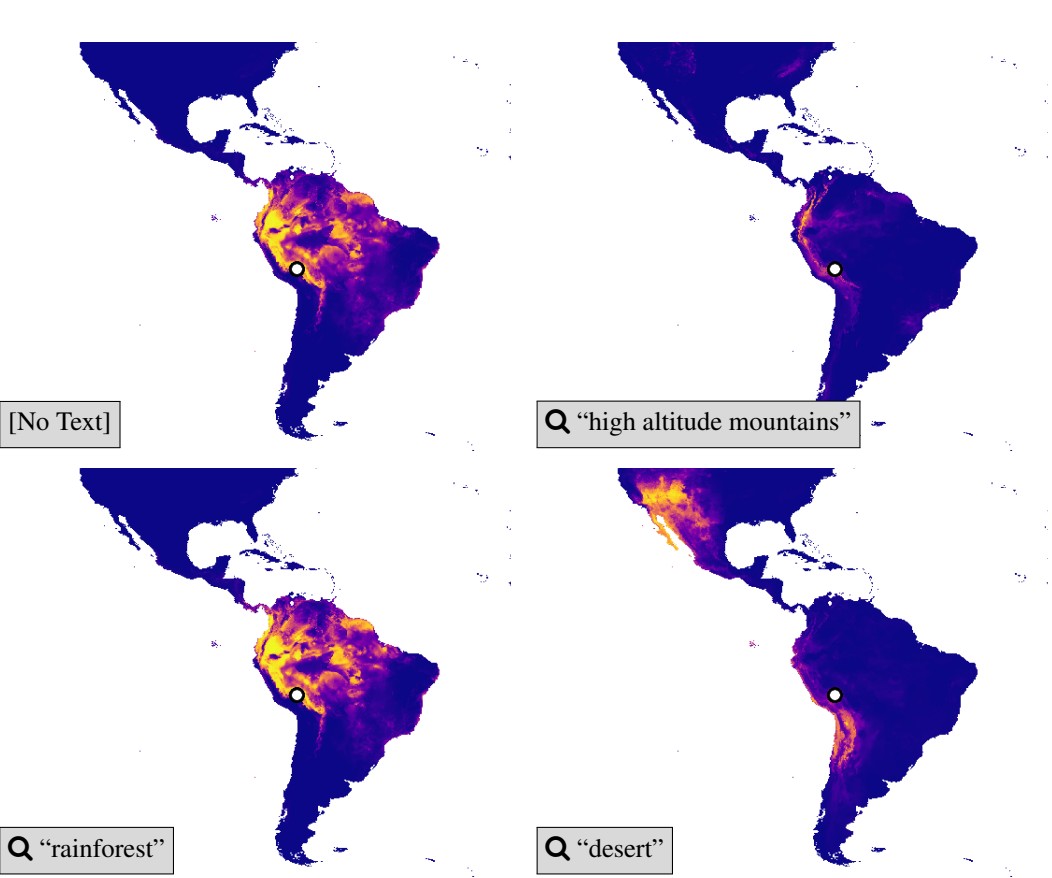

Figure 5: **Controlling range predictions using a single context location with different text**. Given the same single context location, denoted as '○', FS-SINR can generate significantly different range predictions depending on the text provided. This example illustrates a use case where a user may have limited observations but some additional knowledge that can be encoded via text regarding the type of habitat a species of interest could be found in.

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

# Appendix

## A   IMPLEMENTATION DETAILS

### A.1   MODEL ARCHITECTURE

Our FS-SINR architecture consists of four components: The location encoder, $f$; the text encoder, $t$; the transformer encoder, $e$; and the species decoder, $s$. These components comprise of 6,311,680 learnable parameters in total. All non-linearities in FS-SINR are ReLUs.

The location encoder, $f$, is identical to the the one used in Hamilton et al. (2024), which is taken from (Cole et al., 2023). It is composed of an initial linear layer and ReLU nonlinearity followed by four residual layers, where each is a two layer fully connected network with residual connections (He et al., 2015) between the input and output of each residual layer. Each layer contains 256 neurons, and there are 527,616 learnable parameters in total.

The text encoder, $t$, follows the structure of text-based species encoder from Hamilton et al. (2024). In $t$, a pretrained and frozen large language model, GritLM (Muennighoff et al., 2024), is used to produce a fixed 4,096 length embedding from input text. This is then passes through a smaller network to reduce the dimensionality to 256. This smaller network comprises of two residual layers with a hidden layer size of 512. In total, the text encoder contains 3,410,432 learnable parameters.

The transformer encoder, $e$, takes in an arbitrary length set of unordered 256 dimensional tokens produced by $f$ and $t$ as well as two learned tokens that are added to each set of inputs. The "CLS", class, token produces the species range, and a "Register" token, inspired by (Darcet et al., 2024), acts as an additional repository of global information during encoding. Element-wise addition between each token and one of four learned 256 dimensional "token type embeddings" is performed to allow the model to differentiate between tokens from different sources. The transformer itself is composed of four transformer encoder layers, implemented using PyTorch's `nn.TransformerEncoderLayer` (Paszke et al., 2019), based on (Vaswani et al., 2017). Key-Query-Value multi-head attention is used with two "heads". The feed forward components contain 512 neurons per layer, while the token dimensionality is 256. Layer norm is used in each layer, using a default epsilon value of 1e-5 for enhanced numerical stability. In total, $e$ contains 2,176,256 learnable parameters.

Finally the species decoder, $s$, is a simple fully connected network with two hidden layers. Each layer contains 256 neurons, and in total the decoder contains 197,376 learnable parameters.

## A.2 TRAINING

For all training we use the Adam optimizer (Kingma & Ba, 2015) with a learning rate of 0.0005, and an exponential learning rate scheduler with a learning rate decay of 0.98 per epoch, and we use a batch size of 2048. Our training data comes from Cole et al. (2023), comprising of 35.5 million species observations with locations, covering 47,375 species observed prior to 2022 on the iNaturalist platform. However, we remove all species that are found in our evaluation datasets, leaving us with 44,181 species in our training set.

Training comprises of two steps. First, the location encoder, $f$, is trained. This follows the training procedure of Cole et al. (2023) using the $\mathcal{L}_{\text{AN-full}}$ loss function with the positive weighting, $\lambda$, set to 2,048, training for 20 epochs with a dropout of 0.5. To reduce training time without significantly impacting performance we only train on a maximum of 1,000 examples per-species, as done in Cole et al. (2023). Thus our training dataset contains 13.8 million location observations. Secondly, we train all components of FS-SINR, except the pretrained large language model, using our $\mathcal{L}_{\text{AN-full-b}}$ loss with $\lambda$ set to 2,048. We train the location encoder, $f$, again as this improves performance compared to freezing it, seen in Fig. A4. For this part of training we use a dropout of 0.05. We further reduce the training data used to a maximum of 100 examples per-species, leaving 4.0 million training examples, which again increases training speed without a significant impact on performance, as seen in Fig. A5.

Each instance in the training set is used once per epoch as a training example to compute the loss. The training example is not passed through the transformer encoder, $e$, and so does not contribute to making the species embedding vector produced by this part of the model. Instead, additional context information is provided to produce the species embedding. With a 0.7 probability this context information is comprised of 20 context locations and a section of text describing the target species. With 0.2 probability, only the 20 context locations are provided, and with 0.1 probability only the section of text is provided to the model. These context locations are taken from the training data for the target species. As such, a single instance from the training set can be used multiple times per epoch, once as a training example, and potentially many times as a context point. The impact of different distributions of context information provided during training is shown in Fig. A2.

For the text inputs required during this stage of training, we use the text dataset from (Hamilton et al., 2024) comprising of multiple sections of Wikipedia articles for each species in the train set where these are available. This dataset contains 127,484 sections from 37,889 species' articles. Note, that not all 44,181 train species have text data available. When text is not available during training and we are trying to provide both text and context locations to the model, we merely ignore the text and only provide the context locations. When we are attempting to provide just text as context, we instead skip that training example. In practice, during training, we pass all text sections through the frozen large language model once and then store the embeddings produced to use in the current training run and all future runs. This prevents us having to repeatedly query the frozen but resource intensive large language model during training. Training takes approximately ten hours on a single NVIDIA A6000 GPU, requiring about six gigabytes of RAM.

## A.3 BASELINES

We compare our approach to LE-SINR (Hamilton et al., 2024) and SINR (Cole et al., 2023). We follow the original architecture and training procedure for LE-SINR and SINR, with the exception that we enforce that SINR, like LE-SINR and our approach, is trained on our reduced set of 44,181 species which do not include evaluation species.

We also follow the original evaluation procedure for LE-SINR. For few-shot evaluation without text, logistic regression with L2 regularization is performed with location features as input using the few positive examples provided alongside a set of pseudo-negatives drawn half from a uniform random distribution and half from the training data distribution. The regularization weight is set to 20. For text-based "Zero-shot" evaluation we directly make use of the output of the text encoder with the dot product between this and location features giving us a probability of species presence. For few-shot

evaluation, when text is provided, we again perform logistic regression, but the output of the text encoder is used as the "target" that the weights are drawn towards in a modified L2 regularization term. See (Hamilton et al., 2024) for more details. The regularization weight is again set to 20.

The original SINR implementation requires all evaluation species to be part of the training set. We match the adaptations from Hamilton et al. (2024) to allow evaluation on unseen species. After training we remove the learned species heads and keep only the location encoder. During evaluation we perform logistic regression with L2 regularization using location features as input. The regularization weight is again set to 20, and the same method of selecting pseudo-negatives as above is used.

### A.4 EVALUATION

We perform three runs for each experiment using different seeds and report the mean. We display the standard deviation as error bars in our figures. For all evaluations across SINR, LE-SINR, and FS-SINR, the same set of context locations are used for a given species, and these context locations are accessed in the same order, so all evaluations using five context locations are performed with the same five points, and four of those points are those used for evaluations using four context locations, etc. In our few-shot setting, we use at most 50 context locations during both training and evaluation.

## B ABLATIONS

Here we present results from investigating a variety of elements of our FS-SINR model and training procedure. We present plots on a "Symlog" scale, where a linear scale is used between 0 and 1 in order to allow us to show zero-shot results alongside few-shot results. We show a mean of three runs with standard deviations shown as error bars. We also present just the mean values alongside in order to allow easier reading.

### B.1 ABLATING TRAINING CONTEXT LOCATIONS

In Fig. A1 we show "Range Text" evaluation performance on the IUCN dataset for FS-SINR models trained using different amounts of context information. We see that generally increasing the context used during training improves performance, and that having a fixed number of context locations is also beneficial.

### B.2 ABLATING CONTEXT INFORMATION

In Fig. A2 we show "Range Text" evaluation performance on the IUCN dataset for FS-SINR models trained using different combinations of text and location context information during training. We see that good text only zero-shot performance requires sometimes providing just text as context information during training. This forces the model to learn to produce ranges from only text information. Models that are sometimes provided with both text and locations for the same training examples perform best as the number of provided context locations increases. We also see that models trained without text can perform on par with those that see text during training when enough context locations are provided (5 - 10). As we might expect, models that are provided with token types they have not seen during training perform poorly.

### B.3 ABLATING INPUT FEATURES

In Tab. A1 we provide additional zero-shot results expanding on those in Tab. 1 in the main paper. Specifically, we add comparisons to using a different location encoder (i.e., SATCLIP (Klemmer et al., 2023) instead of SINR) and comparisons to using the environmental covariates as in SINR (Cole et al., 2023) that contain information about the location climate in addition to location coordinates.

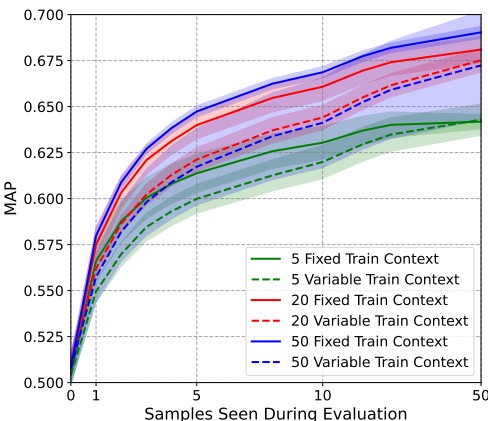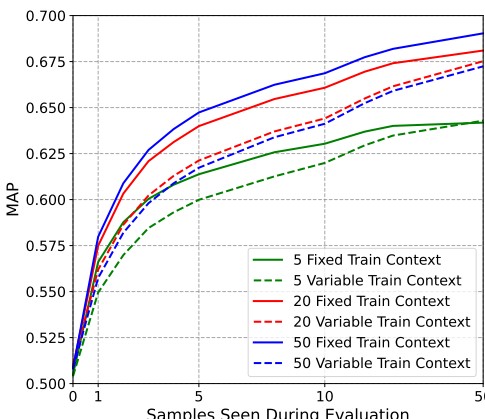

Figure A1: **Impact of amount of train context locations.** Here we evaluate FS-SINR models trained using different amounts of location context locations. Results are shown with standard deviations from three runs (left), and without (right) for clarity. Evaluation is performed with "Range Text" on the IUCN dataset. "Fixed" indicates the same number of context locations were provided for every training example. "Variable" indicates that a uniform random distribution of context locations up to the specified number were provided with each training example. We see that "Variable" generally underperforms compared to "Fixed" and that increasing the train context length tends to increase evaluation performance.

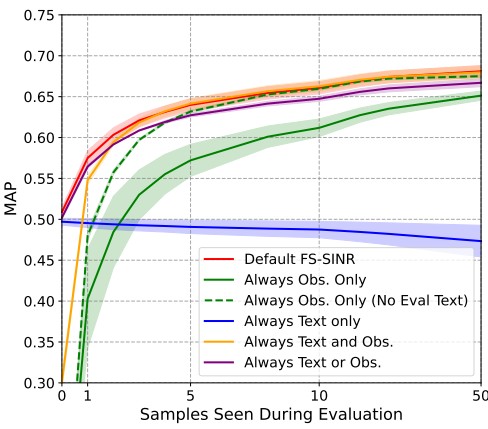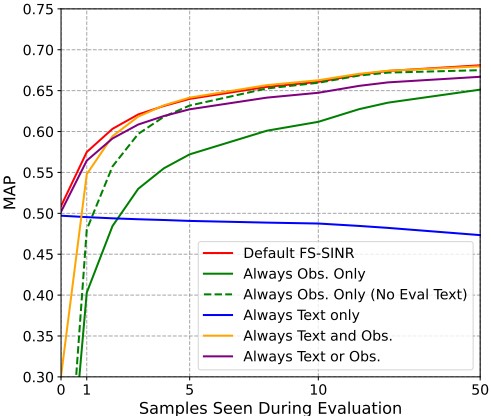

Figure A2: **Impact of train context information.** Here we evaluate FS-SINR models trained using different context information on the IUCN dataset. Results are shown with standard deviations from three runs (left), and without (right) for clarity. Evaluation is performed with "Range Text" unless "No Eval Text" is specified, in which case just locations are provided during eval. 70% of training examples for "Default FS-SINR" provide both location and text context, 20% provide just locations 10% and provide just text. "Always Obs. Only" has only seen locations during training. "Always Text Only" has only seen Text during training. "Always Text and Obs" is always provided with both locations and text during training. "Always Text or Obs." is provided with just locations for 90% of training examples, and just text for the remaining 10%.

## B.4 Ablating Location Encoder

In Fig. A3 we vary the number of datapoints used to pretrain the SINR encoder used in FS-SINR. For both FS-SINR and the SINR baseline, we generally observe that more data is better, and for SINR approaches we see that pretraining the encoder is much better than randomly initializing it. We also show results for a SINR model trained on evaluation species as well as train species. As we saw in Tab. 1 for FS-SINR, the impact on performance is fairly small as these models do not have

Table A1: **Zero-shot results**. We compare to SINR (Cole et al., 2023) and LE-SINR (Hamilton et al., 2024) where no location information is provided to each model. We denote additional meta-data used by models as RT for 'Range Text', HT for 'Habitat Text', and EN for models that use additional environmental covariates from (Cole et al., 2023) as input. TST represents 'Test Species in Train', indicating that a model uses observations for the evaluation species at training time, un-like other models where they are excluded. SATCLIP denotes a variant of our model whereby the SINR encoders are replaced with the image derived location encoders from (Klemmer et al., 2023). Results are presented as MAP, where higher is better.

| Method | Variant | IUCN | S&T |
|---|---|---|---|
| FS-SINR | HT, SATCLIP | 0.20 | 0.43 |
| FS-SINR | RT, SATCLIP | 0.33 | 0.55 |
| SINR | EN, TST | 0.76 | 0.81 |
| FS-SINR | HT, EN, TST | 0.38 | 0.61 |
| FS-SINR | RT, EN, TST | 0.57 | 0.67 |
| FS-SINR | EN | 0.07 | 0.64 |
| LE-SINR | HT, EN | 0.31 | 0.52 |
| FS-SINR | HT, EN | 0.32 | 0.53 |
| LE-SINR | RT, EN | 0.51 | 0.61 |
| FS-SINR | RT, EN | 0.51 | 0.65 |

weights associated with individual species. Unlike the zero-shot SINR model also shown in Tab. 1, our few-shot approach discards these weights and so much of the information learned during training is lost. Due to this we see that our zero-shot performance for SINR models trained on evaluation species is much greater than our few-shot performance with a small number of samples.

In Fig. A4 we also investigate the impact of changing the location encoder entirely. We see that replacing our SINR location encoder with a pretrained, frozen "Satclip" location encoder (Klemmer et al., 2023) significantly harms performance. This may be due to this model being frozen and trained on tasks that do not completely match ours. In comparison a randomly initialised and untrained SINR backbone performs almost identically well as one that has seen a small amount of training data (10 examples per-species in the train set). We also investigate removing the learned location encoder with a simple form of fourier feature encoding (Tancik et al., 2020). In this setting, a pretrained and finetuned SINR type location encoder is still used to encode inputs to the species vector, $w_s$, after it has been produced by the transformer encoder and species decoder, but this model is not used for inputs to the transformer itself. Using these 2 different encoders performs increasingly poorly as the amount of context information increases.

## B.5 ABLATING TRAINING DATA

In Fig. A5 we vary the number of examples per-species that are provided during training. The impact of this is fairly small, with models trained on an intermediate amount of data performing best. We find that the a model trained on only 10 examples per-species performs significantly worse, though it is likely that some of this performance drop is that we must also train this model using 10 context locations per training example rather than the 20 used for the other models, as there is simply not enough data to provide more context information.

## B.6 ABLATING FS-SINR ARCHITECTURE

In Fig. A6 we vary the underlying FS-SINR architecture. Removing several components has a very small effect on model performance, with the removal of the species decoder actually improving results when range text is provided. However, as several ablations perform very similarly, it is difficult to tease out the how much of this effect is due to variance. It is clear however that removing the learnable token type embeddings causes the model to completely fail to learn during training.

In Fig. A7 we show further ablations based around removing the learned location encoder for inputs to the transformer and replacing it with the simple fourier feature encoding also seen in Fig. A4.

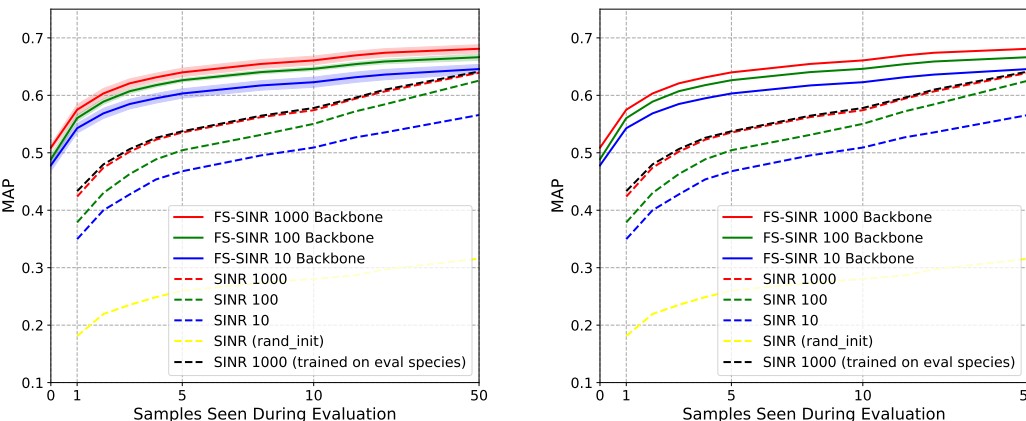

Figure A3: **Impact of Location Encoder Training.** Here we evaluate the performance of SINR and FS-SINR models when the size of the training dataset for the SINR backbone is varied. Results for FS-SINR models are shown with standard deviations from three runs (left), and without (right) for clarity. Evaluations on FS-SINR are performed with "Range Text", while SINR can only make use of location data. "1000", "100", "10" represent the maximum number of examples per class the SINR backbone was trained on. "SINR (rand_init)" is initialized with random weights and is not trained. "(trained on eval species)" means the model was trained on all training and evaluation classes.

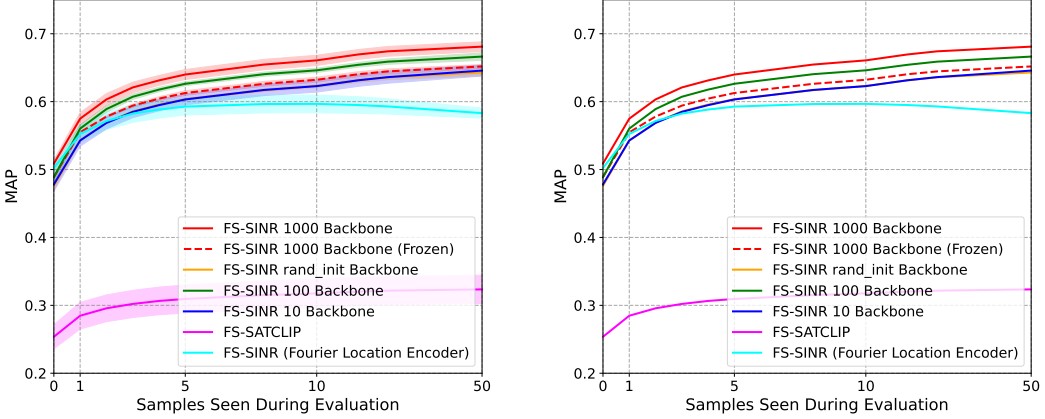

Figure A4: **Impact of Location Encoder.** Here we evaluate the performance of FS-SINR type models with different location encoders. Results are shown with standard deviations from three runs (left), and without (right) for clarity. Evaluation is performed with "Range Text" on the IUCN dataset. "1000", "100", "10" represent the maximum number of examples per class the SINR backbone was trained on. "(Frozen)" indicates that the location encoder parameters were not updated during FS-SINR training. "FS-SATCLIP" replaces the SINR location encoder with a pretrained, frozen location encoder from Klemmer et al. (2023). "FS-SINR (Fourier Location Encoder)" uses the simple fourier feature encoding (Tancik et al., 2020) used in Mildenhall et al. (2021) to match the 256D outputs of the SINR location encoders. These outputs are used directly as inputs to the transformer encoder. After a species token is produced in this way, it is attached to a pretrained and finetuned SINR backbone to produce a range.

When this is removed, other ablations seem to further harm performance, though results for these ablations vary wildly between runs.

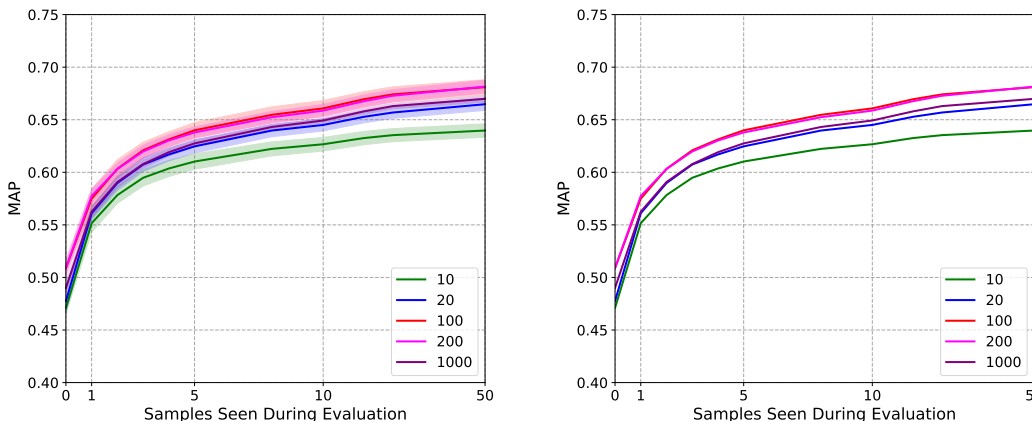

Figure A5: **Impact of Training Data.** Here we evaluate FS-SINR models trained with different amounts of data. Results are shown with standard deviations from three runs (left), and without (right) for clarity. Evaluation is performed with "Range Text" on the IUCN dataset. The labels show the maximum number of examples per-species that FS-SINR is trained on. We see that training on an intermediate amount of training data leads to best performance.

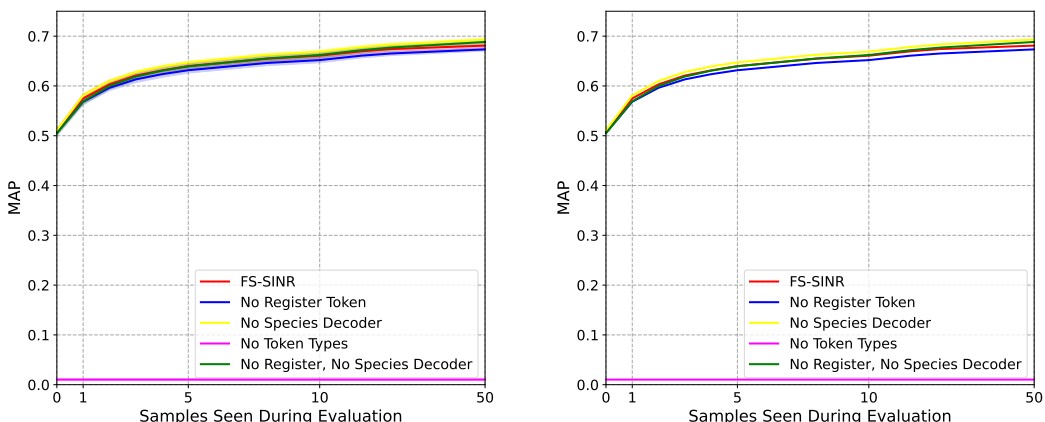

Figure A6: **Ablating model architecture components.** Here we evaluate the performance of FS-SINR type models as we ablate various design choices. Results are shown with standard deviations from three runs (left), and without (right) for clarity. Evaluation is performed with "Range Text" on the IUCN dataset. We see only small changes in performance when removing the register token and the species decoder. However removing the learned token type embeddings has a large impact.

## B.7 TAXONOMIC UNDERSTANDING

Here we investigate the impact of providing FS-SINR with an understanding of taxonomy. For this we provide "Taxonomic Rank Text" (TRT) instead of the Wikipedia-based free-form descriptions of a species that are used for our standard FS-SINR approach. This text gives the taxonomy of the species in decreasing taxonomic rank, in the form "class order family genus species", so for a dog we would give the text "`Mammalia Carnivora Canidae Canis Familiaris`". During training we select a rank uniformly at random and remove all ranks underneath that. We hope that this process will force the model to learn an understanding of the distributions of not only individual species, but also genera, families, etc.. This may be helpful when facing unseen species as knowledge of the genus or family may provide clues about where this species may be found. This is similar to the approach used by LD-SDM (Sastry et al., 2023).

In Tab. A2 we show zero-shot performance for FS-SINR models trained on TRT on the IUCN and S&T evaluation tasks. We see that as we provide additional taxonomic information zero-shot

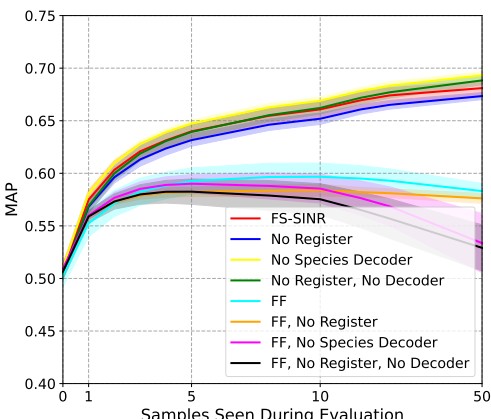 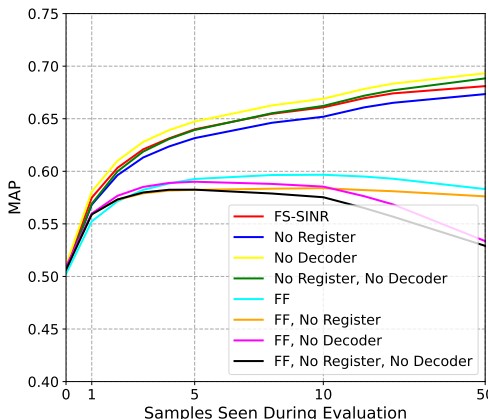

Figure A7: **Further ablating model components.** Here we evaluate the performance of FS-SINR type models as we ablate more components. Results are shown with standard deviations from three runs (left), and without (right) for clarity. Evaluation is performed with "Range Text" on the IUCN dataset. "FF" indicates that the model does not use a SINR backbone to encode location inputs to the transformer encoder. Instead a simple Fourier feature encoding (Tancik et al., 2020) used in Mildenhall et al. (2021) is used to increase the dimensionality of location data to match the token dimension of the transformer encoder. These are used directly as inputs to the transformer encoder. After a species token is produced in this way, it is attached to a standard SINR backbone to produce a range. Removing the SINR backbone for encoding inputs to the transformer has a large impact on performance, especially when more context locations are supplied, and makes the model more sensitive to the impact of other ablations.

performance improves, though it is still much worse than using habitat or range text. This implies that the model has managed to develop some understanding of the distributions of genera etc. and can use this to help it map a novel species that shares higher order taxonomy with species in the training set. In Fig. A8 we show few-shot results for FS-SINR models trained on TRT on the IUCN and SNT evaluation datasets. Zero-shot improvement with increasing taxonomic information is clear, but after very few provided locations this effect seems to disappear.

In Fig. A9 we provide some qualitative zero-shot and few-shot results showing the impact of training on taxonomic text. We see that the model appears to narrow down on the correct range as more specific taxonomy is revealed to it, from predicting across the entire globe when just the class `Aves` is provided, to removing northern latitudes as the family `Columbidae` is added, and finally removing the new world when the genus is provided. This broadly matches the actual distribution of these taxonomic ranks.

Table A2: **Zero-shot results with taxonomy rank text**. We denote additional metadata used by models as RT for 'Range Text' and HT for 'Habitat Text'. 'Species', 'Genus', 'Family', 'Order', 'Class' refer to models trained and evaluated using taxonomic rank text. Taxonomic information up to and including the specified rank is provided during evaluation.

| Method | Variant | IUCN | S&T |
|--------|---------|------|-----|
| FS-SINR |  | 0.05 | 0.18 |
| FS-SINR | HT | 0.33 | 0.53 |
| FS-SINR | RT | 0.52 | 0.64 |
| FS-SINR | Class | 0.05 | 0.19 |
| FS-SINR | Order | 0.06 | 0.20 |
| FS-SINR | Family | 0.12 | 0.25 |
| FS-SINR | Genus | 0.18 | 0.30 |
| FS-SINR | Species | 0.21 | 0.34 |

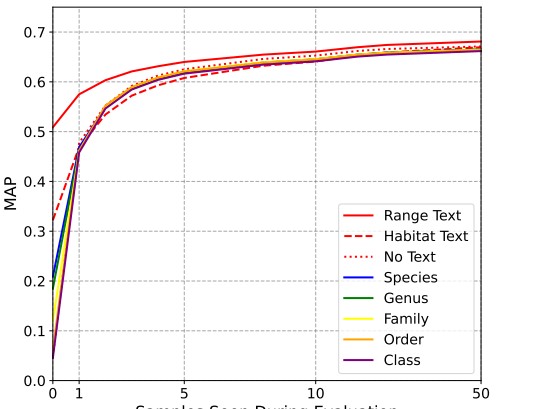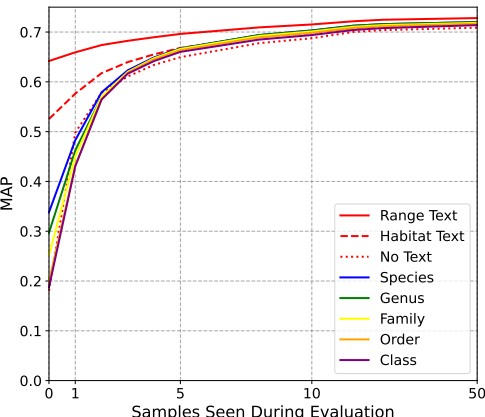

Figure A8: **Impact of training and evaluating with Taxonomic Rank Text.** Here we evaluate FS-SINR models trained using different context information on the IUCN dataset (left), and the S&T dataset (right). "Class" indicates that only the taxonomic class of the species is provided as text during evaluation. "Order" indicates that the taxonomic class followed by the order is provided as a text string during evaluation, and so on, such that "Species" indicates that a text string in the format "class order family genus species" is provided during evaluation. Providing more specific taxonomic text increases zero-shot performance. This is also presented Tab. A2. However we see that even the full taxonomy does not provide as much signal as habitat and range text for zero-shot range mapping. These more detailed texts provide more useful information for zero-shot range mapping - either actually mentioning geographic locations in the case of range text, or allowing the model to narrow predictions down to areas with specific features such as mountains and forests in the case of habitat text. When a single context location is provided, the choice of taxonomy text no longer seems to impact performance at all. It is possible that training on these less informative tokens means the model learns to pay less "attention" to these text tokens compared to the Wikipedia-based text tokens usually used during training. This could explain why different rank taxonomy text tokens seemingly provide no benefit when any context locations are provided to the model.

## C  ADDITIONAL QUALITATIVE RESULTS

In this section we provide additional qualitative results.

### C.1  QUALITATIVE RESULTS

As in LE-SINR Hamilton et al. (2024), by jointly training on text and locations, FS-SINR is able to spatially ground abstract non-species concepts in a zero-shot manner. In Fig. A10 we see some examples where different text concepts, that are very different from the species range or habitat text provided during training, are grounded in sensible locations on the map. In Fig. A11 we compare models with and without text cues. As we increase the number of context locations, the two different models converge to more similar range predictions. In Fig. A12 we provide another example similar to Fig. 5 in the main paper. Here, we again fix the context location and show the impact of changing the text. We can see that different text prompts can result in quite different predicted ranges. In Figs. A13 and A14 we visualize the model range predictions for two different species when richer habitat or range text is provided. We observe that the combination of text and context locations (here only location is provided) results in the best performance. In Fig. A15 we visualize FS-SINR range predictions for the `Yellow-footed Green Pigeon` for models that have had different random initializations (i.e., different random seeds). We observe that there is a relatively large amount of variance in the outputs produced given the same input data.

### C.2  VISUALIZING EMBEDDINGS

In Fig. A16 we show Independent Component Analysis (ICA) derived projections of the location encoder features for FS-SINR, LE-SINR, and SINR approaches. We encode locations around the

world into 256 dimensional representations by passing them through location encoders from our models, and we then reduce these to three dimensions and show them as RGB colors as in (Cole et al., 2023). We also show this for an FS-SINR model trained on taxonomic rank text. Locations with similar colors should have similar location features and represent locations that the model thinks may share species. Across all models higher frequency changes in location features are seen in areas where we have more training data. This can be seen particularly clearly by comparing the United States and Europe versus central Asia or Africa.

## C.3   QUALITATIVE COMPARISONS

Here we present qualitative comparisons of the ranges produced by FS-SINR, LE-SINR, and SINR. In Fig. A17 we show range estimates for the `Brown-banded Watersnake`, using range text for FS-SINR and LE-SINR approaches. In Fig. A18 we show range estimates for the `Brown-headed Honeyeater`, using habitat text for FS-SINR and LE-SINR approaches. Finally in Fig. A19 we show range estimates for the `Crevice Swift`, without providing text. Overall, SINR produces more diffuse ranges and requires more samples to narrow down the range. LE-SINR and FS-SINR appear to have very different zero-shot behaviours, with LE-SINR frequently seeming to predict presence in almost no locations at all, while FS-SINR tends to produce a zero-shot range that is too large.

## D   ADDITIONAL QUANTITATIVE RESULTS

In this section we present additional quantitative results. We include results from Fig. 3 in Tab. A3 and Tab. A4, for IUCN and S&T evaluations respectively.

Table A3: **IUCN zero-shot and few-shot results.** Here we present IUCN evaluation results for the models shown in Fig. 3 in tabular form. SINR and LE-SINR without text cannot produce a range map without at least one context point. Results are presented as MAP, where higher is better.

| | FS-SINR | | | LE-SINR | | | SINR |
|---|---|---|---|---|---|---|---|
| # Context | Range | Habitat | No Text | Range | Habitat | No Text | No Text |
| 0 | 0.52 | 0.33 | 0.05 | 0.48 | 0.28 | - | - |
| 1 | 0.57 | 0.47 | 0.48 | 0.55 | 0.48 | 0.47 | 0.42 |
| 2 | 0.60 | 0.54 | 0.56 | 0.57 | 0.53 | 0.52 | 0.47 |
| 3 | 0.62 | 0.57 | 0.60 | 0.58 | 0.55 | 0.54 | 0.50 |
| 4 | 0.63 | 0.59 | 0.62 | 0.59 | 0.57 | 0.56 | 0.52 |
| 5 | 0.64 | 0.61 | 0.63 | 0.60 | 0.58 | 0.57 | 0.54 |
| 8 | 0.65 | 0.63 | 0.65 | 0.61 | 0.60 | 0.59 | 0.56 |
| 10 | 0.66 | 0.64 | 0.66 | 0.62 | 0.61 | 0.60 | 0.57 |
| 15 | 0.67 | 0.66 | 0.67 | 0.63 | 0.63 | 0.62 | 0.59 |
| 20 | 0.67 | 0.66 | 0.67 | 0.64 | 0.64 | 0.63 | 0.61 |
| 50 | 0.68 | 0.67 | 0.67 | 0.66 | 0.66 | 0.66 | 0.64 |

## D.1   RESULTS BY REGION

Here we show the average false positive error by location on the IUCN evaluation dataset.

In Fig. A20 we show the average false positive error for zero-shot range estimation using text for FS-SINR and LE-SINR, alongside the distribution of data in our training set. It appears that training data density is somewhat negatively correlated with error. We observe that LE-SINR has significantly lower false positive error globally, however Tab. 1 shows that MAP is also lower. Appendix C.3 shows that LE-SINR tends to predict very small areas for zero-shot range mapping, which explains both the lower false positive error and the lower MAP.

In Fig. A21 we show the average false positive error for FS-SINR for few-shot range estimation.

## D.2   RESULTS BY SPECIES RANGE SIZE

In this section we show plots indicating the average MAP for species in our IUCN evaluation dataset, separated by range size. We include standard deviation error bars. Unlike earlier plots, these are not

Table A4: **S&T zero-shot and few-shot results.** Here we present S&T evaluation results for the models shown in Fig. 3 in tabular form. SINR and LE-SINR without text cannot produce a range map without at least one context point. Results are presented as MAP, where higher is better.

| # Context | FS-SINR | | | LE-SINR | | | SINR |
| --- | --- | --- | --- | --- | --- | --- | --- |
| | Range | Habitat | No Text | Range | Habitat | No Text | No Text |
| 0 | 0.64 | 0.53 | 0.18 | 0.60 | 0.52 | - | - |
| 1 | 0.66 | 0.58 | 0.50 | 0.64 | 0.60 | 0.52 | 0.49 |
| 2 | 0.67 | 0.62 | 0.58 | 0.66 | 0.62 | 0.57 | 0.55 |
| 3 | 0.68 | 0.64 | 0.61 | 0.67 | 0.64 | 0.60 | 0.58 |
| 4 | 0.69 | 0.66 | 0.64 | 0.67 | 0.65 | 0.61 | 0.59 |
| 5 | 0.70 | 0.67 | 0.65 | 0.68 | 0.66 | 0.62 | 0.60 |
| 8 | 0.71 | 0.69 | 0.68 | 0.69 | 0.67 | 0.65 | 0.63 |
| 10 | 0.72 | 0.70 | 0.69 | 0.69 | 0.68 | 0.66 | 0.64 |
| 15 | 0.72 | 0.71 | 0.70 | 0.70 | 0.69 | 0.68 | 0.67 |
| 20 | 0.72 | 0.71 | 0.71 | 0.71 | 0.70 | 0.69 | 0.68 |
| 50 | 0.73 | 0.72 | 0.71 | 0.73 | 0.72 | 0.72 | 0.72 |

generated from the results of three runs, but from the differences in performance between individual species within a range size group.

In Fig. A22 we break down performance of zero-shot approaches by range size for both FS-SINR and LE-SINR. In Fig. A23 we break down performance of low-shot approaches by range size for both FS-SINR and LE-SINR, when provided with habitat text. Finally in Fig. A24 we break down performance of low-shot approaches where no text is provided for FS-SINR, LE-SINR, and SINR.

We find that for all models and settings, performance varies very strongly with range size. This is most significant in the zero-shot setting. FS-SINR performs well compared to our baselines, though all models struggle with very small ranges. We also see that performance worsens for the very largest ranges.

### D.3 RESULTS BY TAXONOMIC CLASS

In this section we break down results on the IUCN evaluation dataset by taxonomic class. We include standard deviation error bars. Unlike earlier plots, these are not generated from the results of three runs, but from the differences in performance between individual species within a taxonomic class. Four taxonomic classes are present in this dataset, namely `Amphibia`, `Aves`, `Mammalia`, and `Reptilia`.

In Fig. A25 we display zero-shot performance for FS-SINR and LE-SINR using range and habitat text. We observe that `Aves` and especially `Mammalia` outperform the other classes, particularly when habitat text is provided. Albert et al. (2018) suggest that of the 20 most 'charismatic' species in the western world, all but the `Great White Shark` and `Crocodile` are mammals, and Trimble & Van Aarde (2010) show that scientific research is heavily focused on mammals. We may be seeing the impact of this, where mammals are more likely to have detailed wikipedia pages where we drew our textual training and evaluation data from.

In Fig. A26 we investigate how these differences in performance between taxonomic classes change as more location data is provided. We see that for both FS-SINR and LE-SINR, even a single location reduces the gap significantly and after 5 context locations the difference is minimal, though mammals do continue to perform best for a given model and setting.

Finally in Fig. A27 We compare FS-SINR to LE-SINR, SINR with our few-shot modifications as in Hamilton et al. (2024), and SINR trained on evaluation species as in Cole et al. (2023). We see that FS-SINR tends to perform very well across all classes.

### D.4 ALTERNATIVE PERFORMANCE METRICS

Here we provide additional results for the main models from Fig. 3 using a new 'distance weighted' MAP evaluation metric. This is inspired by the evaluation conducted in LD-SDM (Sastry et al., 2023). This metric is based on mean average precision (MAP), however we now weight predictions

by distance from the true range, i.e., predicting the presence of a species far from where it is said to be found is penalized more than predicting the presence of a species in a location that is very close to existing observations, but is still actually outside the range. We intend that this metric more closely aligns with a human's judgment on how 'correct' a range is, compared to standard MAP. By considering both metrics we can be more confident that the improvement in range mapping performance that FS-SINR provides is not just a consequence of how we are measuring the performance.

We determine the weight for location $\boldsymbol{x}$ as

$$w_{\boldsymbol{x}} = 1 + \frac{d_{range}(\boldsymbol{x})}{d_{antipodal}} h, \tag{3}$$

where $d_{range}(\boldsymbol{x})$ is the distance along the earth's surface from point $\boldsymbol{x}$ to the nearest point of the expert-derived range using for evaluation, and $d_{antipodal}$ is the distance along the earth's surface between two points on opposite sides of the earth. While this distance does vary very slightly in different locations as the earth is not a perfect sphere, for this experiment we have set $d_{antipodal}$ to 20,037.5 km. $h$ is the 'distance weight hyperparameter' and determines how much this metric penalizes incorrect predictions far from the range relative to close to the range. The metric is implemented equivalent to scikit-learn's *average_precision_score* sample_weight parameter (Pedregosa et al., 2011). We evaluate performance using the standard 'unweighted MAP', i.e., where $h = 0$ and so we are calculating MAP as usual, and 'distance weighted MAP' with $h = 9$ and $h = 99$. We selected these settings so that errors on the opposite side of earth from the true range are penalized 10 and 100 times more than errors close to the true range.

Results on the IUCN evaluation dataset can be found in Fig. A28. These are from a single run, as opposed to the average of three repeats used for 'unweighted MAP' in Fig. 3. As the weight is increased, we observe a general reduction in overall performance. While there is no change in the relative ordering of different models, and FS-SINR outperforms LE-SINR and SINR across all settings of $h$, we do observe that FS-SINR and LE-SINR models that use habitat text during evaluation seem to decrease in performance more with larger $h$ compared to other approaches. They are likely most effected by the larger weight, as habitat text can cause the model to predict presence in locations around the world with similar habitat features such as mountains, forest, or desert, despite these locations being far from the true range. This appears to be true of both FS-SINR and LE-SINR. For LE-SINR we see that when evaluating using unweighted MAP, using habitat text outperforms not using text, while when we evaluate using weighted MAP, using habitat text performs worse than not using text. In Fig. A29, we display zero-shot results for two species where there is a large difference in performance based on the two metrics. In both cases the language only FS-SINR variant incorrectly predicts the species to be present far from the expert-derived range.

### D.5 ADDITIONAL FEW-SHOT BASELINES

Here we provide additional few-shot baselines based on Prototypical Networks (Snell et al., 2017). Our approach is very similar to Snell et al. (2017) although we use the SINR location encoder of our models as the 'embedding function', allowing us to generate few-shot results for a novel species without any retraining. Using this method, SINR and LE-SINR models can be used to estimate the range of a novel species without requiring training to learn a new species vector.

In order to do this, we first encode our known 'presence' locations using the location encoder of our chosen model and then take an average of these points to generate a 'prototype' for the presence class. We select pseudo-negatives in the same manner as Hamilton et al. (2024) and similarly encode and average these in order to generate a prototype for the 'absent' class.

We represent these prototypes as

$$c_k = \frac{1}{|S_k|} \sum_{\boldsymbol{x}_i \in S_k} f_{\boldsymbol{\theta}}(\boldsymbol{x}_i), \tag{4}$$

where $k \in \{present, absent\}$ indicates the class of the prototype, and $S$ is the 'support set', i.e., the set of locations $\boldsymbol{x}$ that we use to create our prototypes. In our case, $S_{present}$ is the set of locations of the small number of available observations for our target species, while $S_{absent}$ is the set of pseudo-negative locations that we have selected according to Hamilton et al. (2024). $f_{\boldsymbol{\theta}}()$ indicates the location encoder of our model.

To generate a probability of presence or absence at any location $\boldsymbol{x}$, we encode $\boldsymbol{x}$ using our location encoder and calculate the negative squared euclidean distance in our high dimensional 'location encoder space' between $\boldsymbol{x}$ and each prototype. We then use these values as the 'logits' in a softmax function to generate our probabilities. Putting this together, we can calculate the probability of presence as

$$p_{\text{present}}(\boldsymbol{x}) = \frac{e^{-d(f_{\boldsymbol{\theta}}(\boldsymbol{x}), \mathbf{c}_{\text{present}})^2}}{e^{-d(f_{\boldsymbol{\theta}}(\boldsymbol{x}), \mathbf{c}_{\text{present}})^2} + e^{-d(f_{\boldsymbol{\theta}}(\boldsymbol{x}), \mathbf{c}_{\text{absent}})^2}}, \tag{5}$$

where $d(\boldsymbol{a}, \boldsymbol{b})$ represents the euclidean distance between $\boldsymbol{a}$ and $\boldsymbol{b}$.

We present few-shot results using this method in Fig. A30 with certain results from 3 included for comparison. We see that the performance of these 'prototype' approaches is significantly worse than our FS-SINR approach and our learning-based SINR and LE-SINR baselines. On the less discriminative S&T dataset, performance when $|S_k| = 1$ is similar to that of the other methods. However, for both prototype approaches and both evaluation datasets, performance actually decreases as we increase the number of provided context locations.

In Fig. A31 we present qualitative results visualizing the few-shot estimated range for the `Kalahari Scrub-Robin` produced by FS-SINR and by a SINR model using the prototype approach. We see that the estimated range for the 'prototype SINR' becomes significantly worse as we add another context location. We find that averaging location encoder features from multiple 'presence' locations tends to produce a less useful prototype. We attempt to explain this finding next.

The features from a single location may represent ecologically meaningful information about the local environment, and a prototype produced from a single location will have the same representation in location encoder space as the location it is produced from. This suggests that measuring the distance from another encoded location to this prototype may tell us how 'different' the environment of the new location is compared to the location used to create the prototype. This information is helpful for producing an estimate of a species range. However when we have a larger support set and so average the location encoder features of multiple locations, the prototype that is generated may exist in a non-meaningful part of location encoder space, not associated with any real world locations or environmental conditions. The distance between this prototype and an encoded location becomes less indicative of a 'difference' in environment, and so this distance becomes less helpful for estimating the presence of a species at the new location. Therefore as we increase the number of context points which form the 'present' support set, we actually decrease the performance of our prototype approaches.

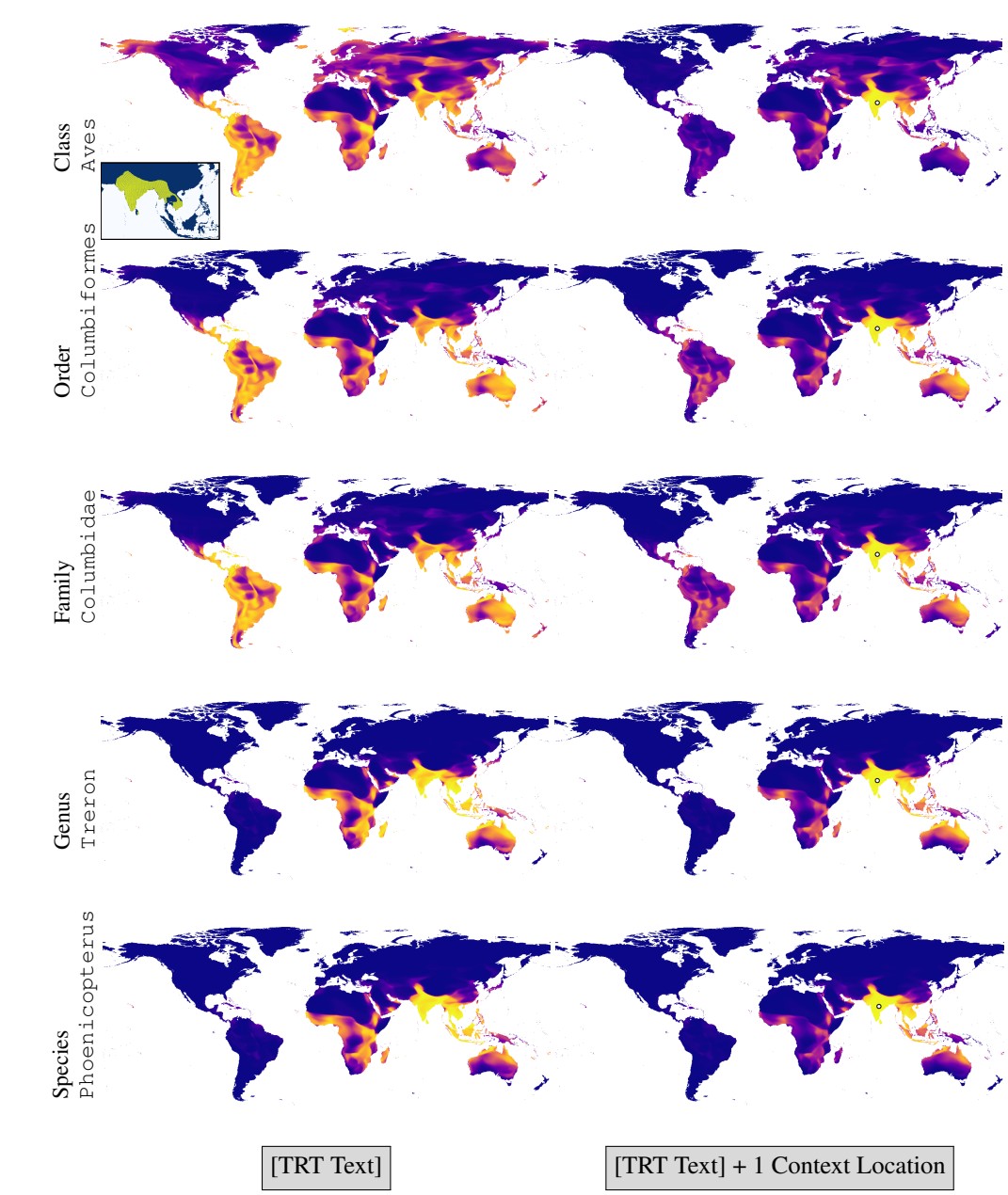

[TRT Text]   [TRT Text] + 1 Context Location

Figure A9: **Zero-shot and one-shot range estimation using Taxonomic Rank Text (TRT).** Here we see range predictions for `Treron Phoenicopterus` or the `Yellow-footed Green Pigeon` from an FS-SINR model trained on taxonomic rank text as in LD-SDM (Sastry et al., 2023), with expert-derived range inset. As seen in Fig. A8 and Tab. A2, The text-based zero-shot predictions seem to more closely match the expert-derived range as more of the taxonomic rank text of the species is provided. Taxonomic rank text allows the model to somewhat localize predictions to areas where species sharing the provided taxonomy ranks are present in the training set. For example, `Aves` or `Birds` are globally distributed and we see the model attempt to output this in the zero-shot 'Class' vizualisation. `Columbiformes` and `Columbidae` or `Pigeons and Doves` are not found in the extreme north and providing these ranks reduces predictions in these areas (and much of the northern hemisphere). The model mostly manages to identify that `Treron` or 'Green Pigeons' are found only in Africa and parts of Asia. A single observation significantly contracts the predicted ranges, particularly when less taxonomic information is provided. Click on taxonomic names to visit the iNaturalist page for that taxonomic rank, where you can see the geographic distribution of observations of that taxa, which may resemble that in our training data.

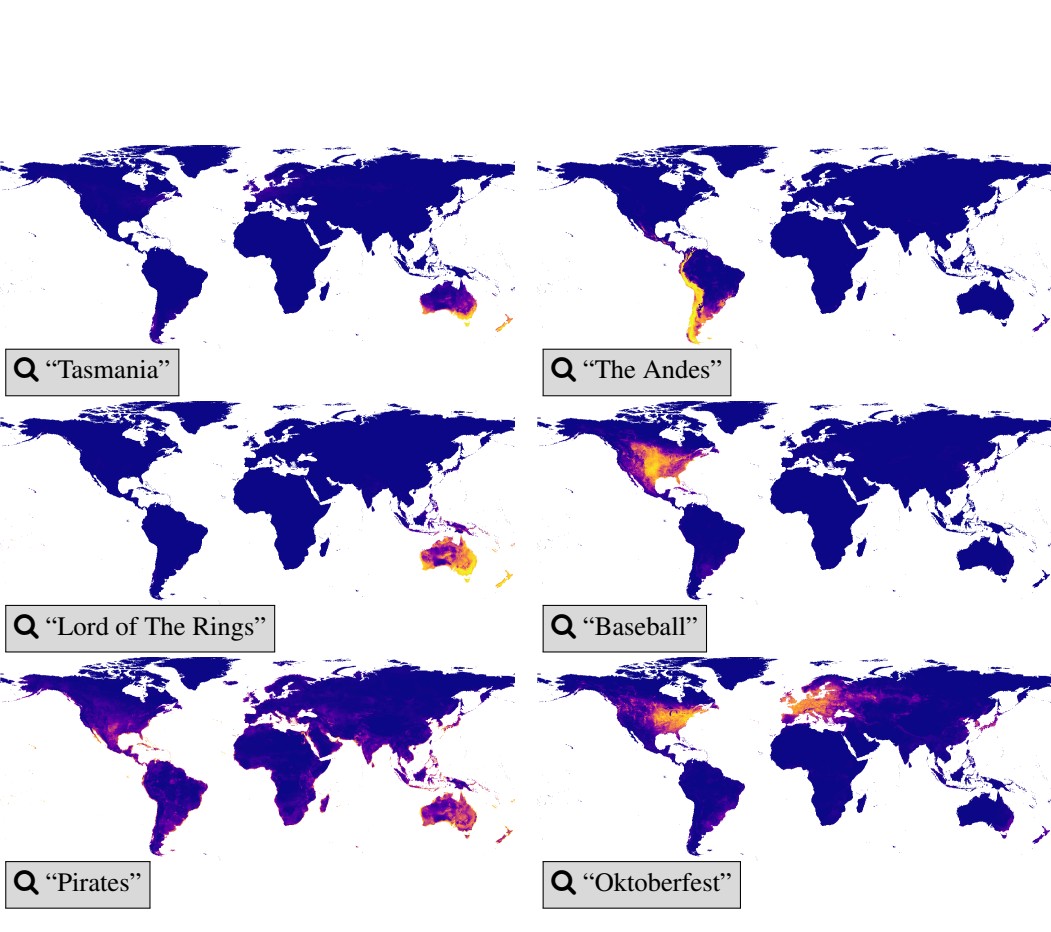

Figure A10: **Zero-shot non-species concepts**. We can evaluate the model in a zero-shot manner using only text information, i.e., without any locations. Here, we observe that FS-SINR, like LE-SINR (Hamilton et al., 2024), can localize abstract concepts in geographic space, despite never being trained to explicitly do so. The model achieves this as it learns to make connections between species text and information already contained in the pretrained language encoder we use. However, we do note failure/ambiguous cases such as the "Pirate" example in the bottom row.

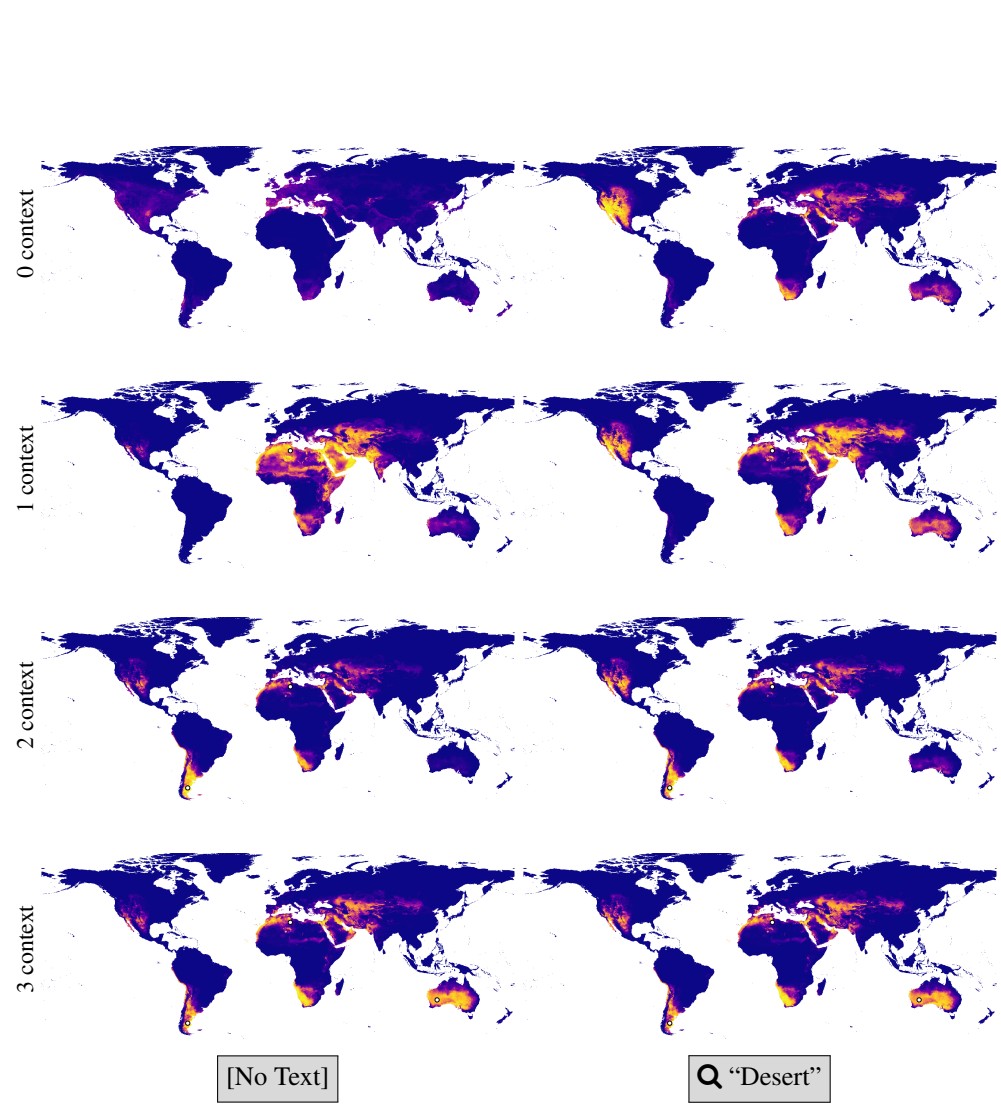

Figure A11: **Varying the context information provided**. Here we change the context information provided to FS-SINR. The model on the left column receives no text input, but the one on the right gets the text "Desert". Additionally, in each row we increase the number of context locations provided, from zero to three, denoted as '∘'. We observe that the model on the right that uses text already has a strong prior about the species being present at desert-like locations, e.g., see first row where no context locations are provided. As soon as one context location is added in North Africa (second row), the model generates a new prediction with an increased probability that the species is present there.

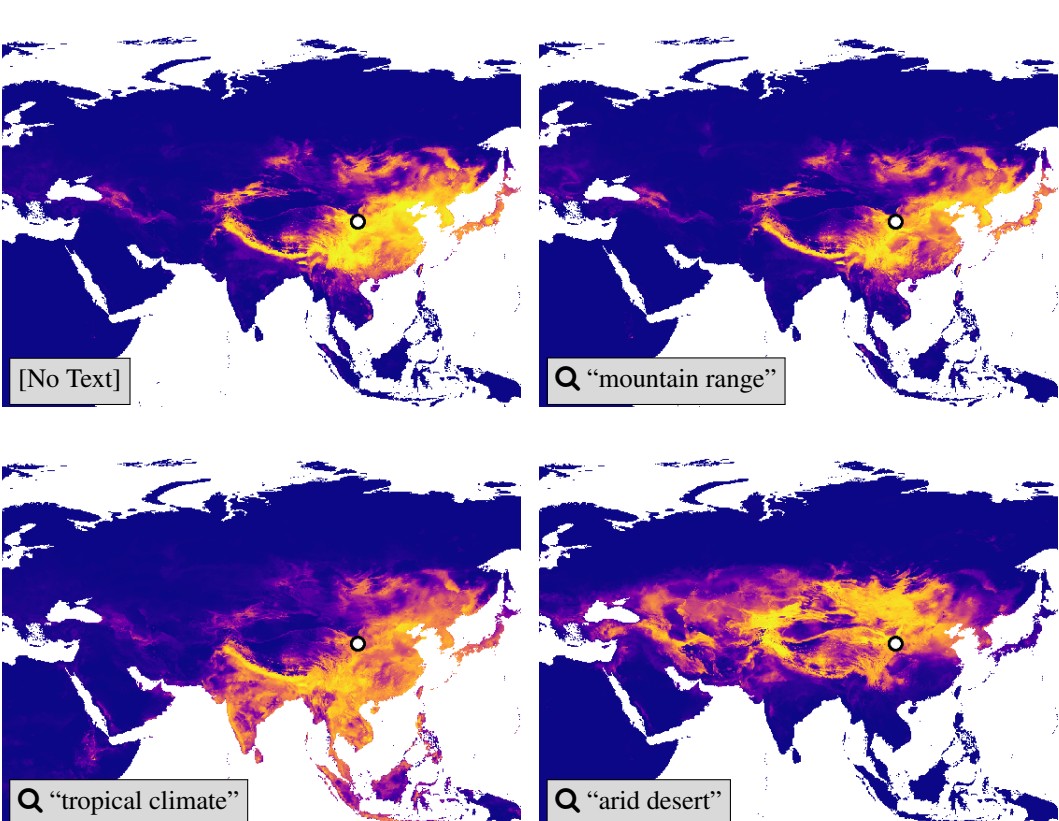

Figure A12: **Controlling range predictions using a single context location and text**. Here we show another example similar to Fig. 5 in the main paper. Given the same context location, denoted as '○', FS-SINR can produce significantly different range predictions depending on the text provided. This example illustrates a use case where a user may have limited observations but some additional knowledge regarding what type of habitat a species of interest could be found in.

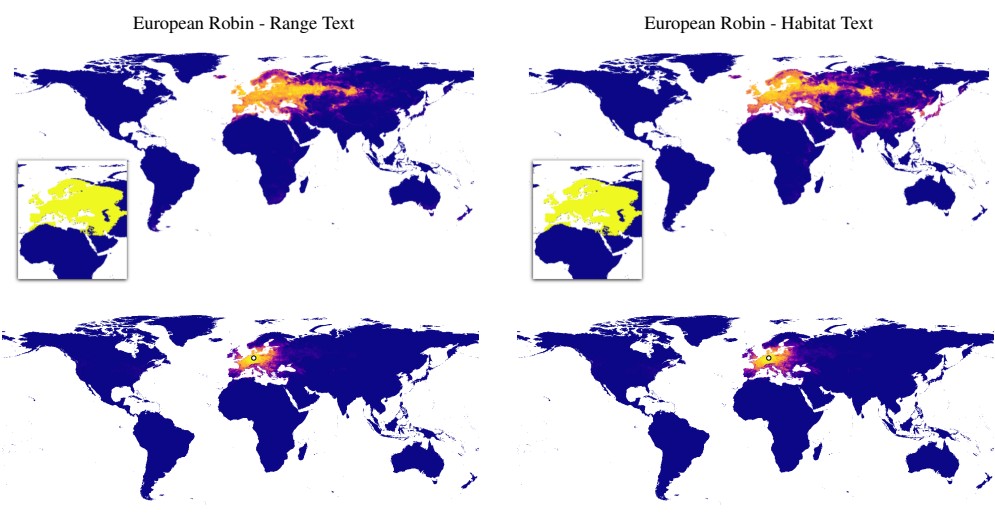

European Robin - Range Text · European Robin - Habitat Text

The European robin is found across Europe, east to Western Siberia and south to North Africa; it is sedentary in most of its range except the far north. It also occurs in the Atlantic islands as far west as the Central Group of the Azores and Madeira. It is a vagrant in Iceland and has been introduced to other regions, including North America and Australia, but these introductions were unsuccessful.

The European robin inhabits a variety of habitats, including gardens, parks, woodlands, and forests. It prefers areas with dense vegetation and is often found near human settlements. It is also found in mountainous regions and can be seen in urban areas, such as cities and towns.

Figure A13: **Using text descriptions.** Here we illustrate the zero-shot (top row) and one-shot (bottom row) range estimations based on text descriptions for the European Robin, using 'Range' (left), and 'Habitat' (right) text, shown below the range estimates. Expert derived range maps are shown inset.

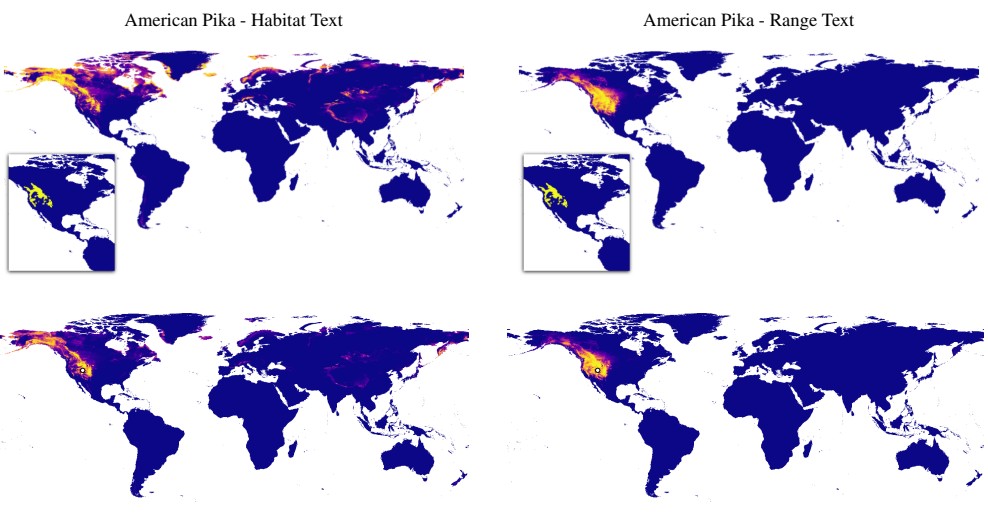

American Pika - Habitat Text · American Pika - Range Text

The American pika (Ochotona princeps) is found in the mountains of western North America, usually in boulder fields at or above the tree line, from central British Columbia and Alberta in Canada to the US states of Oregon, Washington, Idaho, Montana, Wyoming, Colorado, Utah, Nevada, California, and New Mexico.

Pikas inhabit talus fields that are fringed by suitable vegetation in alpine areas. They also live in piles of broken rock. Sometimes, they live in man-made substrate such as mine tailings and piles of scrap lumber. Pikas usually have their den and nest sites below rock, around 20-100 cm (8-39 in) in diameter, but often sit on larger and more prominent rocks.

Figure A14: **Using text descriptions.** Here we illustrate the zero-shot (top row) and one-shot (bottom row) range estimations based on text descriptions for the American Pika, using 'Range' (left), and 'Habitat' (right) text, shown below the range estimates. Expert derived range maps are shown inset.

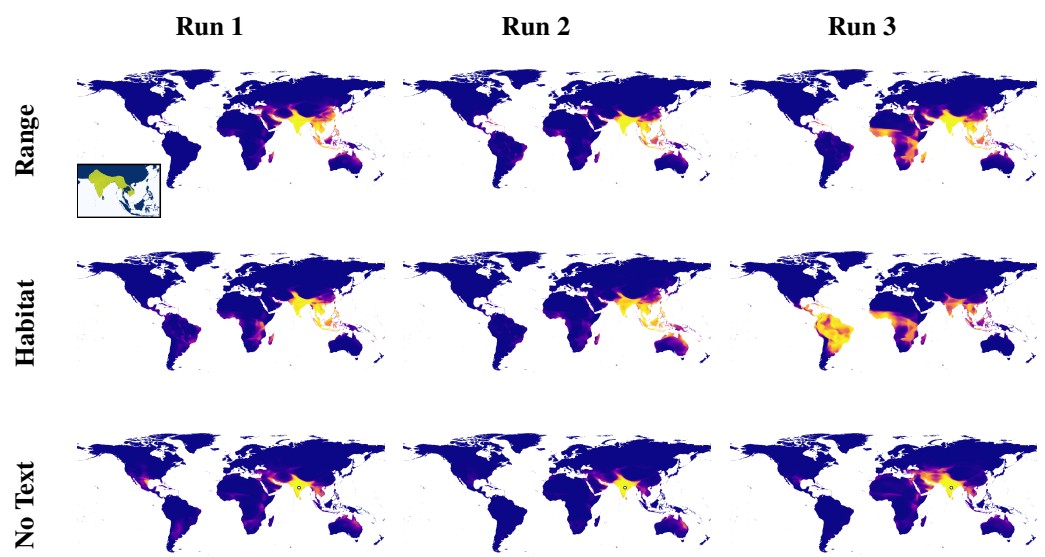

Figure A15: **Impact of random initialization on FS-SINR.** Here we display range estimates for the `Yellow-footed Green Pigeon` from three different FS-SINR models where different random seeds were used to initialize each model during training. We show zero-shot results using 'range text' (top) and 'habitat text' (middle), and also few-shot results using one context location with no text (bottom). The IUCN expert derived range is shown inset. We see that even when provided with the same inputs, different models can perform very differently when this input is very sparse (e.g., just text or one context point). While most of the Indian part of the actual range is included for all input types and runs, there is significant variability across the runs in other geographic areas.

*Range Text:* "The yellow-footed green pigeon is found in the Indian subcontinent and parts of Southeast Asia. It is the state bird of Maharashtra."

*Habitat Text:* "The species is a habitat generalist, preferring dense forest areas with emergent trees, especially Banyan trees, but can also be spotted in natural remnants in urban areas. They forage in flocks and are often seen sunning on the tops of trees in the early morning."

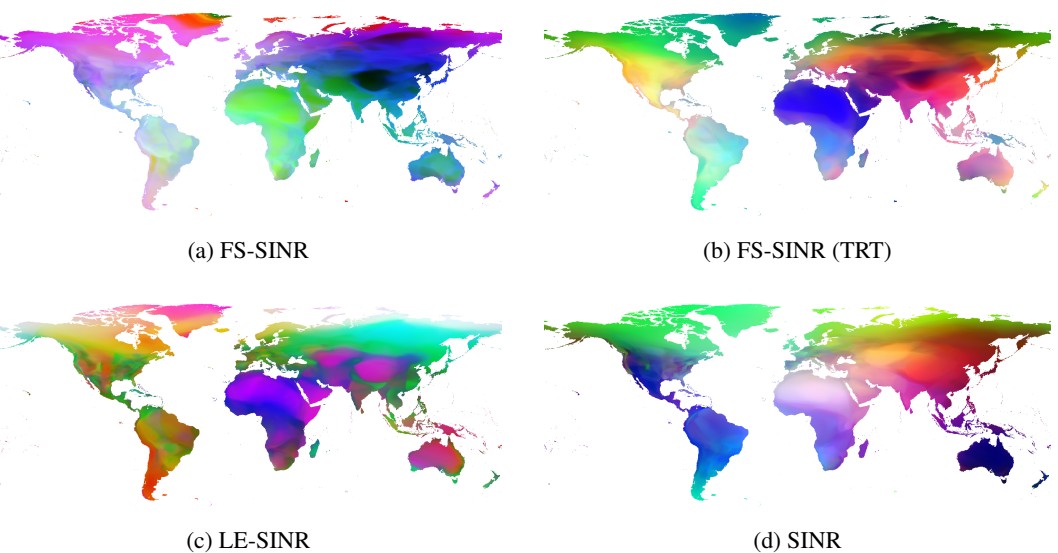

Figure A16: **Visualization of the learned features of different location encoders.** Here we project high dimensional location features down to three dimensions using Independent Component Analysis.

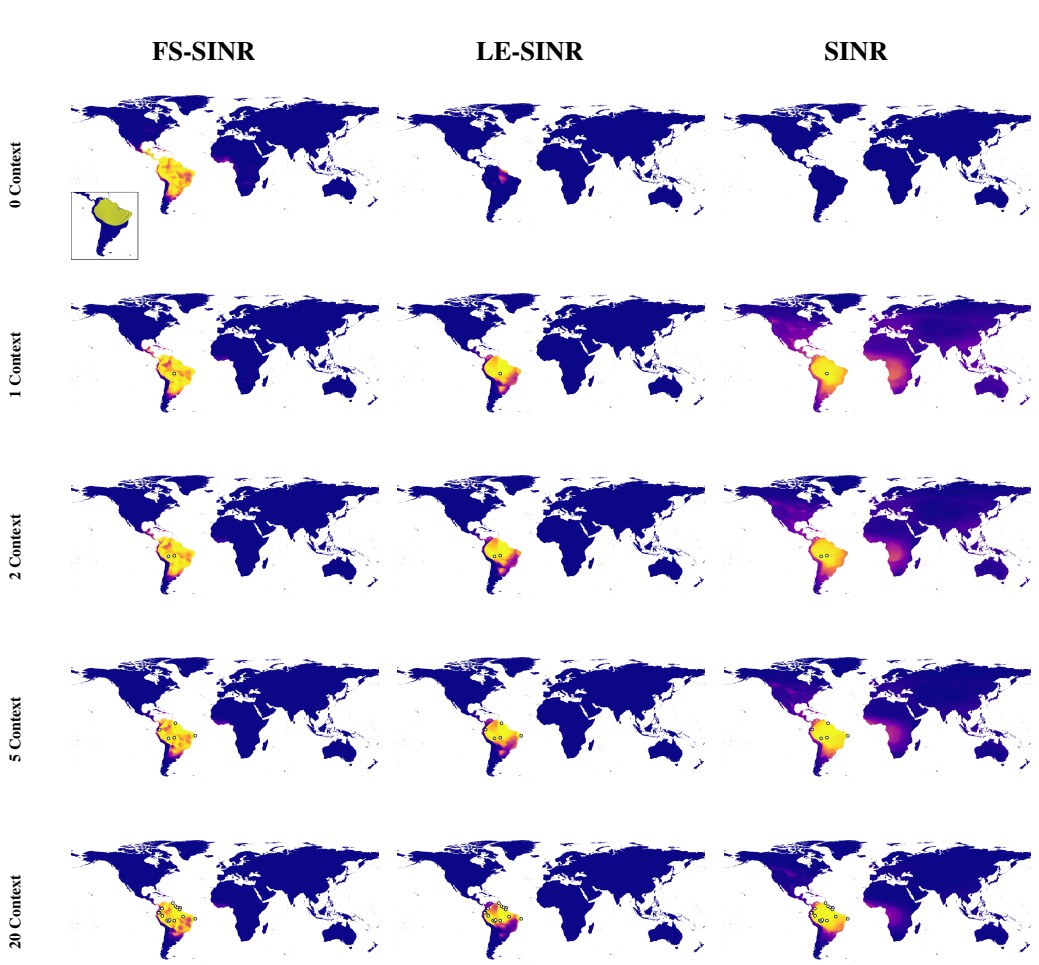

Figure A17: **Comparing estimated ranges across models.** Here we see zero-shot and few-shot range estimates produced by FS-SINR, LE-SINR, and SINR for the `Brown-banded Watersnake`, with expert derived range inset. We provide range text to FS-SINR and LE-SINR as well as context locations, but SINR is not capable of accepting text and so we show a blank map for the zero-shot range estimate. We see that LE-SINR underestimates the range using only text, while FS-SINR overestimates it. SINR requires more location data than the other approaches to localize the range to South America. *Range Text:* "The Brown-banded water snake (Helicops angulatus) is found in tropical South America and Trinidad and Tobago."

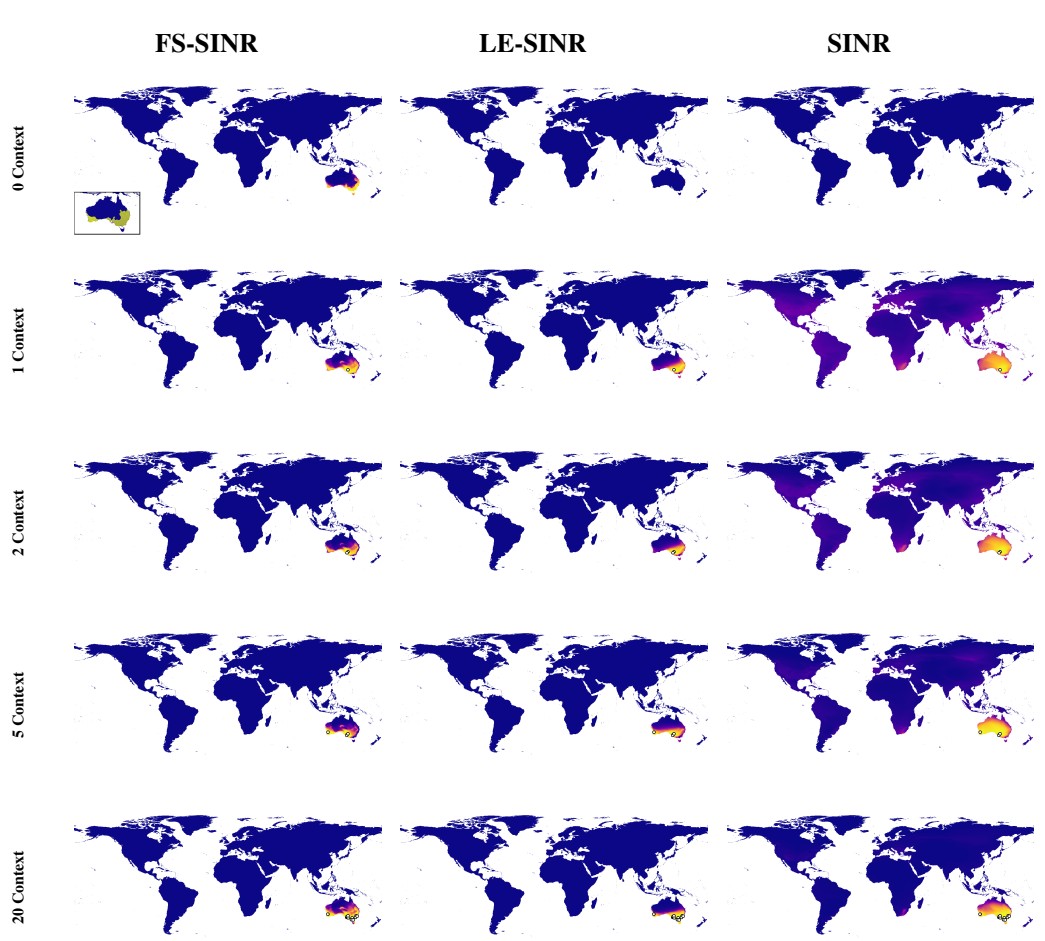

Figure A18: **Comparing estimated ranges across models.** Here we see zero-shot and few-shot range estimates produced by FS-SINR, LE-SINR, and SINR for the `Brown-headed Honeyeater`, with expert derived range inset. We provide habitat text to FS-SINR and LE-SINR as well as context locations, but SINR is not capable of accepting text and so we show a blank map for the zero-shot range estimate. We again see LE-SINR underestimate the range using only text, while FS-SINR has very good zero-shot performance for this species. We see that SINR again requires more location data to narrow down the range and even after 20 locations the range is still significantly larger than the other models, and extends into South Africa. *Habitat Text:* The brown-headed honeyeater inhabits temperate forests and Mediterranean-type shrubby vegetation. It is typically found in tall trees, where it forages by probing in the bark of trunks and branches.

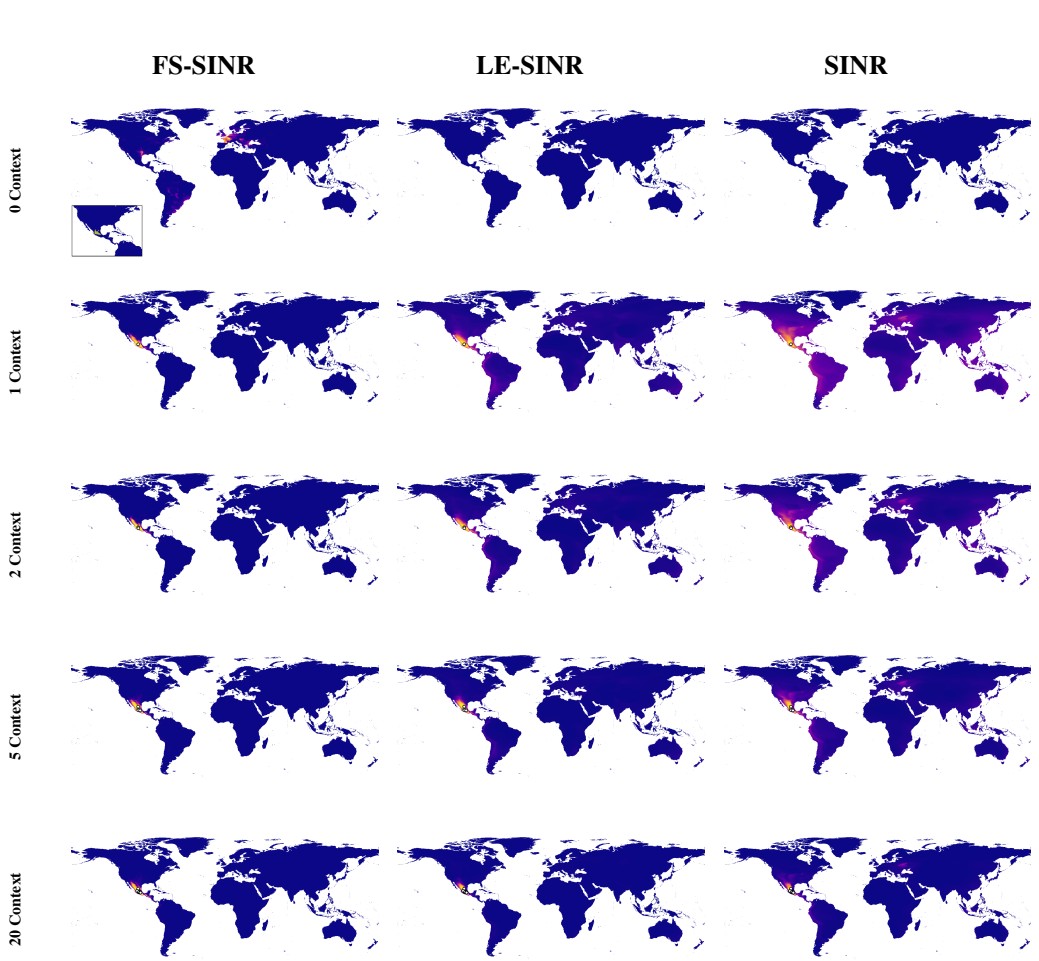

Figure A19: **Comparing estimated ranges across models.** Here we see few-shot range estimates produced by FS-SINR, LE-SINR, and SINR for the `Crevice Swift` lizard, with expert derived range in Mexico inset. No text is provided and so no sensible zero-shot prediction can be made for any model. However while LE-SINR and SINR cannot produce an output for this and so we show a blank map, FS-SINR can generate a predicted range just from feeding the learned CLS and register tokens with no other information into the transformer encoder. The range that is produced is contained wthin the model or the learned tokens itself rather than from any further inputs. We see that it appears to somewhat match the distribution of training data we see in Fig. A20. Absent additional information, the model guides predictions towards areas where it as seen many species during training. This may be an unhelpful bias when attempting to model novel species. SINR again produces more diffuse ranges than the other methods, though all approaches struggle to model these small ranges, as seen in Appendix D.2.

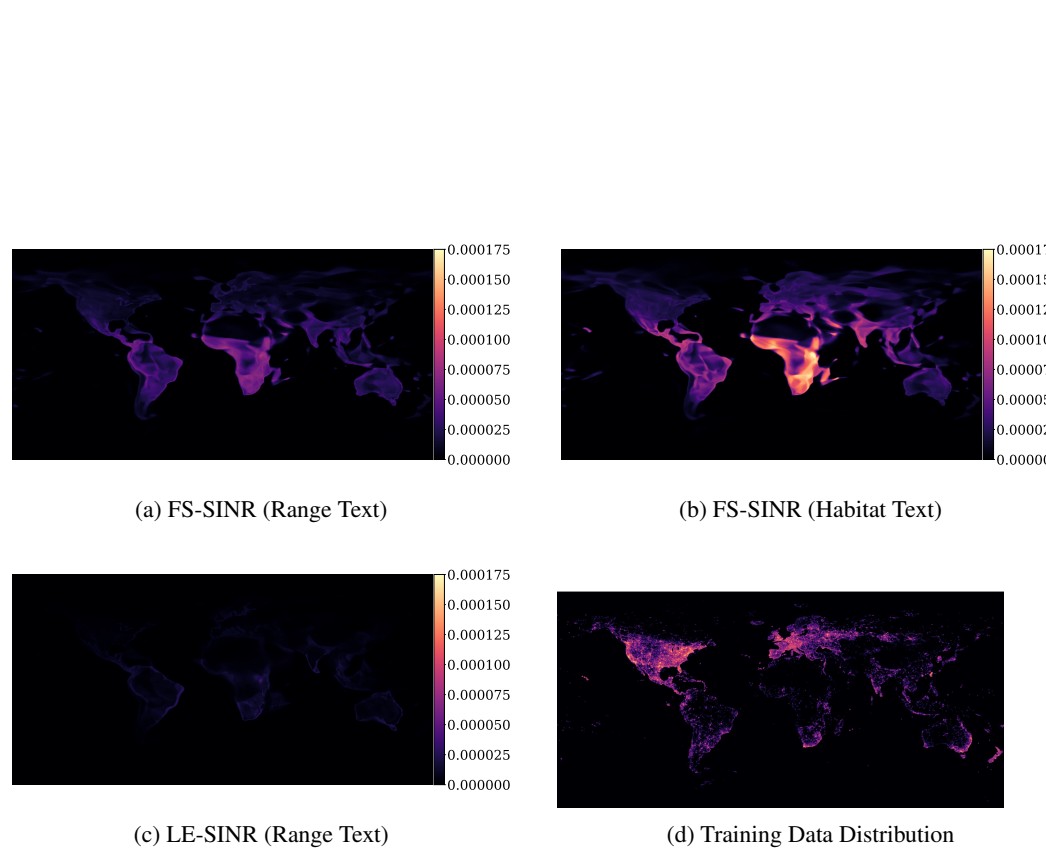

(a) FS-SINR (Range Text)

(b) FS-SINR (Habitat Text)

(c) LE-SINR (Range Text)

(d) Training Data Distribution

Figure A20: **Average false positive error by location for zero-shot approaches.** Here we see the geographic distribution of false positive errors from FS-SINR and LE-SINR models provided with only text during IUCN evaluation. We observe that FS-SINR provided with text (a and b) tends to have fewer errors in areas well covered by our training data such as North America and Europe, while areas with less training data such as Africa have significantly higher error. We also see that LE-SINR (c) has a significantly lower false positive error across all locations than FS-SINR, though quantitative results in Tab. 1 show LE-SINR performs worse at zero-shot range estimation. This suggests that LE-SINR tends to 'underpredict' ranges while FS-SINR is more prone to 'overpredict' ranges. In some areas such as South America, the coast has higher false positive error than inland areas. This is due to model predictions for land based species 'bleeding' slightly into the ocean, and it appears to be an issue for both LE-SINR and FS-SINR. (d) shows the distribution of our geographic training data.

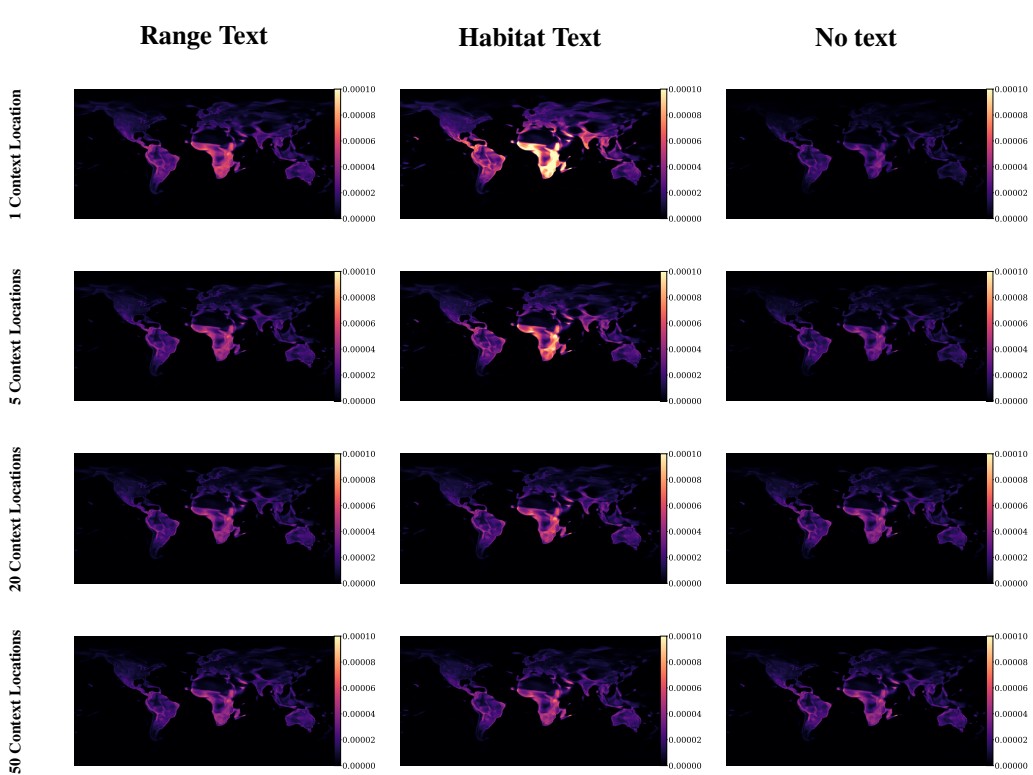

Figure A21: **Average false positive error by location for few-shot approaches.** Here we see average false positive error of FS-SINR on IUCN evaluation. Providing any text leads to an increase in the false positive error, although Fig. 3 suggests that this text still helps with range mapping. As the number of provided context locations increases, the impact of the text is reduced and the distribution of errors appear similar.

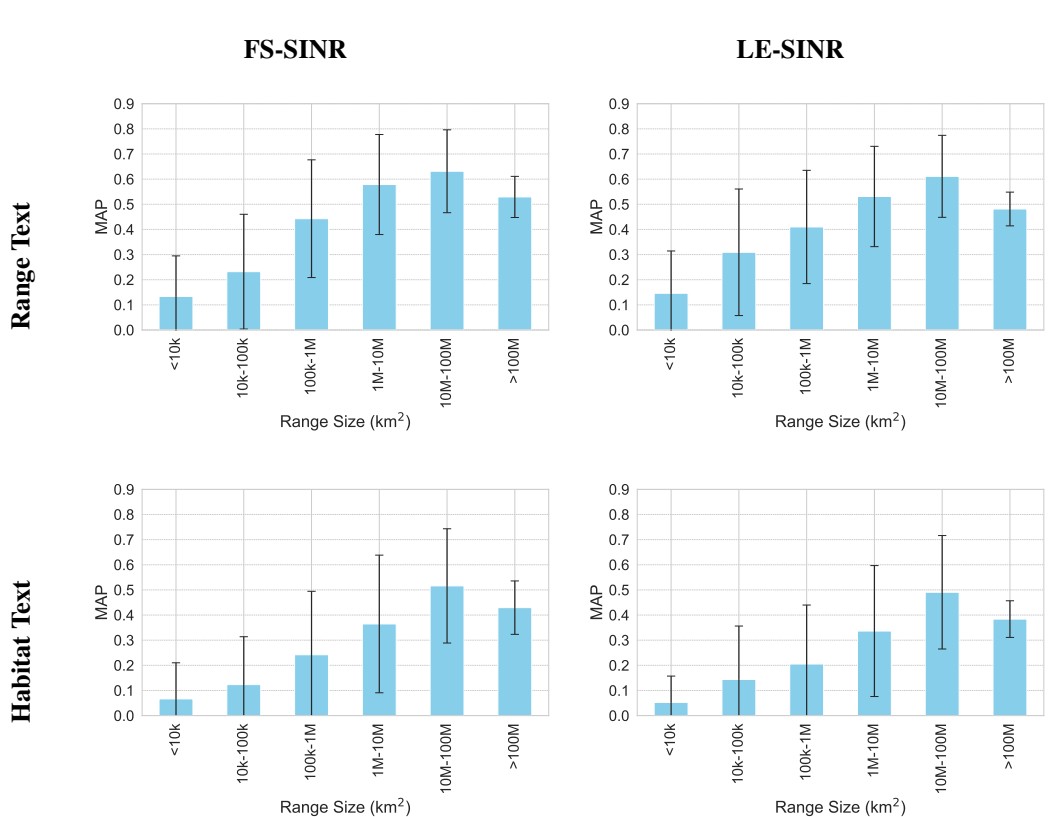

Figure A22: **Zero-shot performance by range size.** Here we see zero-shot IUCN evaluation results grouped by range size for FS-SINR and LE-SINR, using range text and habitat text. We see that for both models, performance is strongly dependent on range size, with ranges between 10 million and 100 million $km^2$ being modelled most succesfully. FS-SINR tends to perform better than LE-SINR for large range species, while LE-SINR tends to perform better for smaller range species.

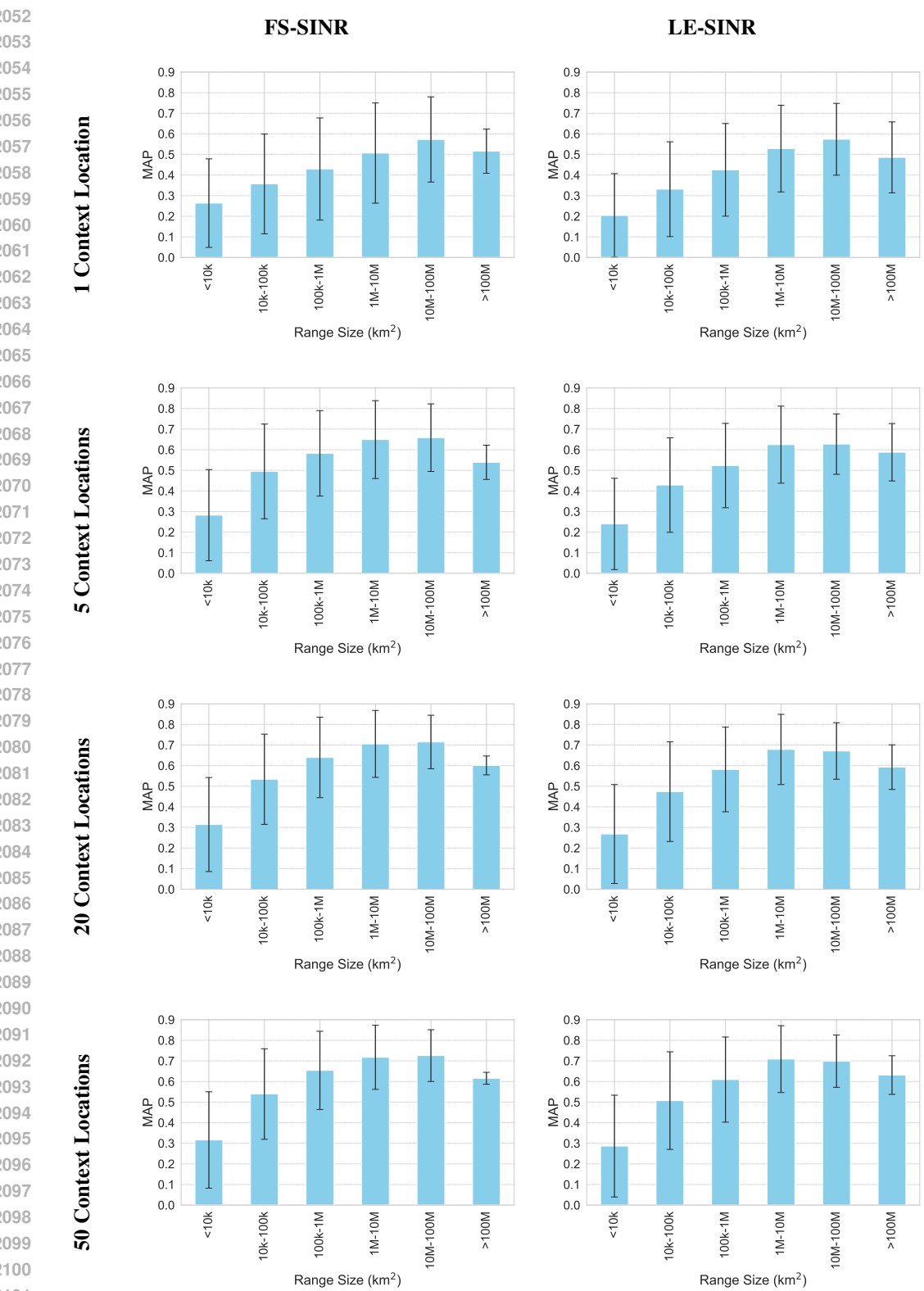

Figure A23: **Few-shot performance by range size.** Here we see few-shot IUCN evaluation results using habitat text for FS-SINR and LE-SINR for a range of context locations. Increasing the number of context locations generally increases performance across both models, though the bias towards intermediate range sizes seen in Fig. A22 remains.

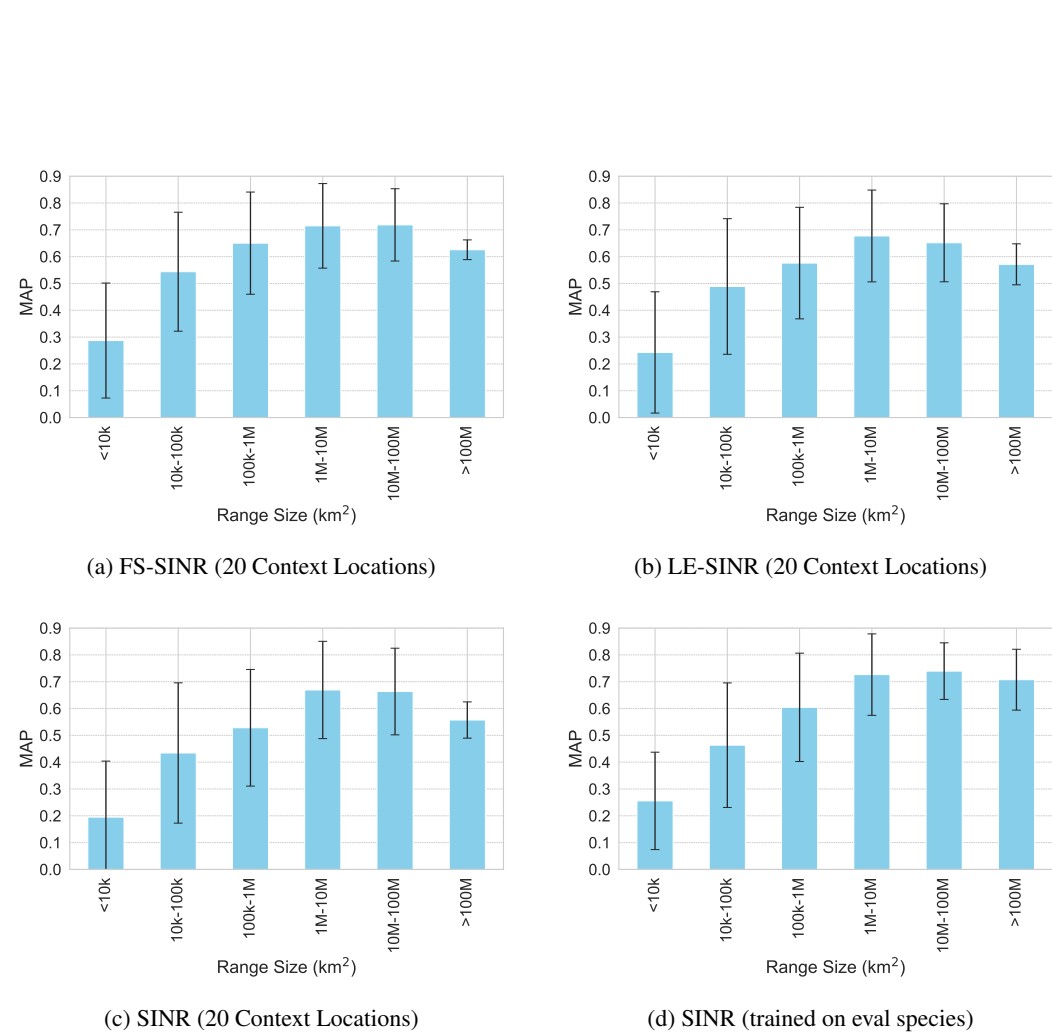

(a) FS-SINR (20 Context Locations)

(b) LE-SINR (20 Context Locations)

(c) SINR (20 Context Locations)

(d) SINR (trained on eval species)

Figure A24: **Few-shot performance by range size without text.** Here we see IUCN evaluation performance for a range of models where text was not provided during evaluation. "SINR (trained on eval species)" was trained on up to 1000 examples per-species for both the train and eval species, and the species vectors learned during training are used during evaluation, as in Cole et al. (2023). Without text, FS-SINR is most capable of modeling small range sizes, though the "trained on eval species" SINR performs best on large ranges.

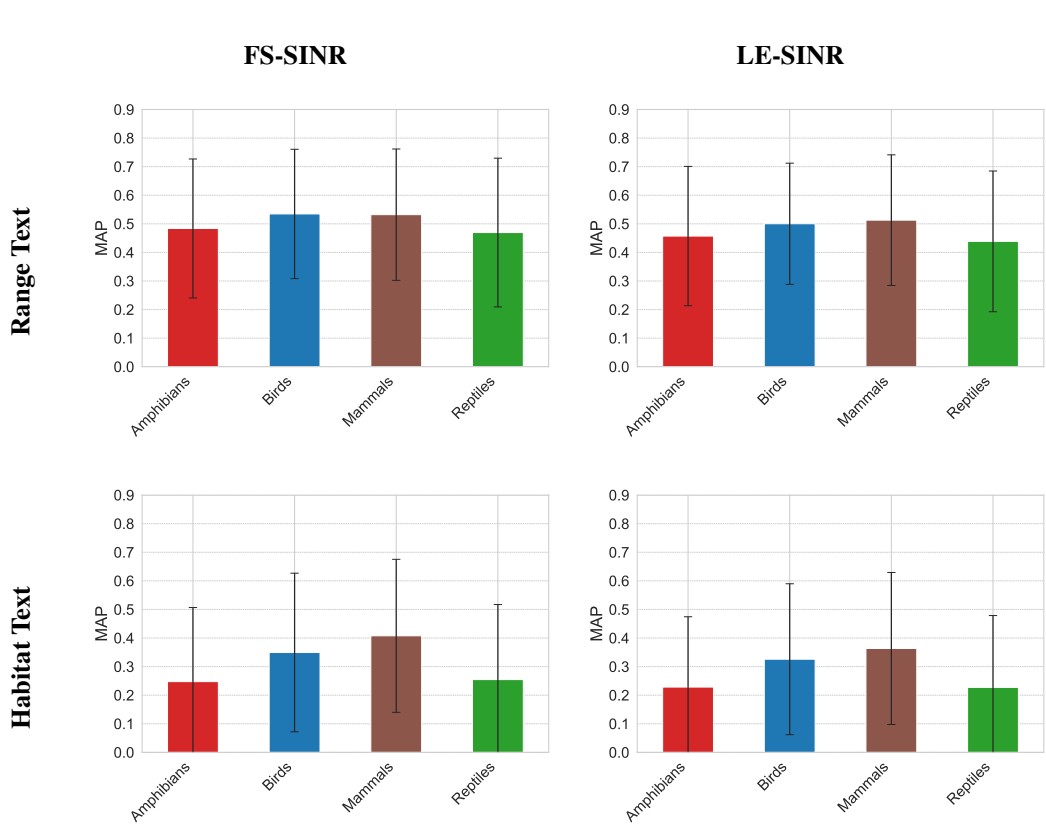

Figure A25: **Zero-shot performance by taxonomic group.** Here we see Zero-shot IUCN evaluation results for FS-SINR and LE-SINR, using range text and habitat text. FS-SINR outperforms LE-SINR across all taxonomic categories. We observe that for both models, birds and mammals outperform amphibians and reptiles. This is particularly pronounced when using habitat text. This may be due to these groups being particularly well studied by researchers and appreciated by people in general, so the text data available for these taxonomic groups tends to be richer and more likely to describe habitat preferences in detail.

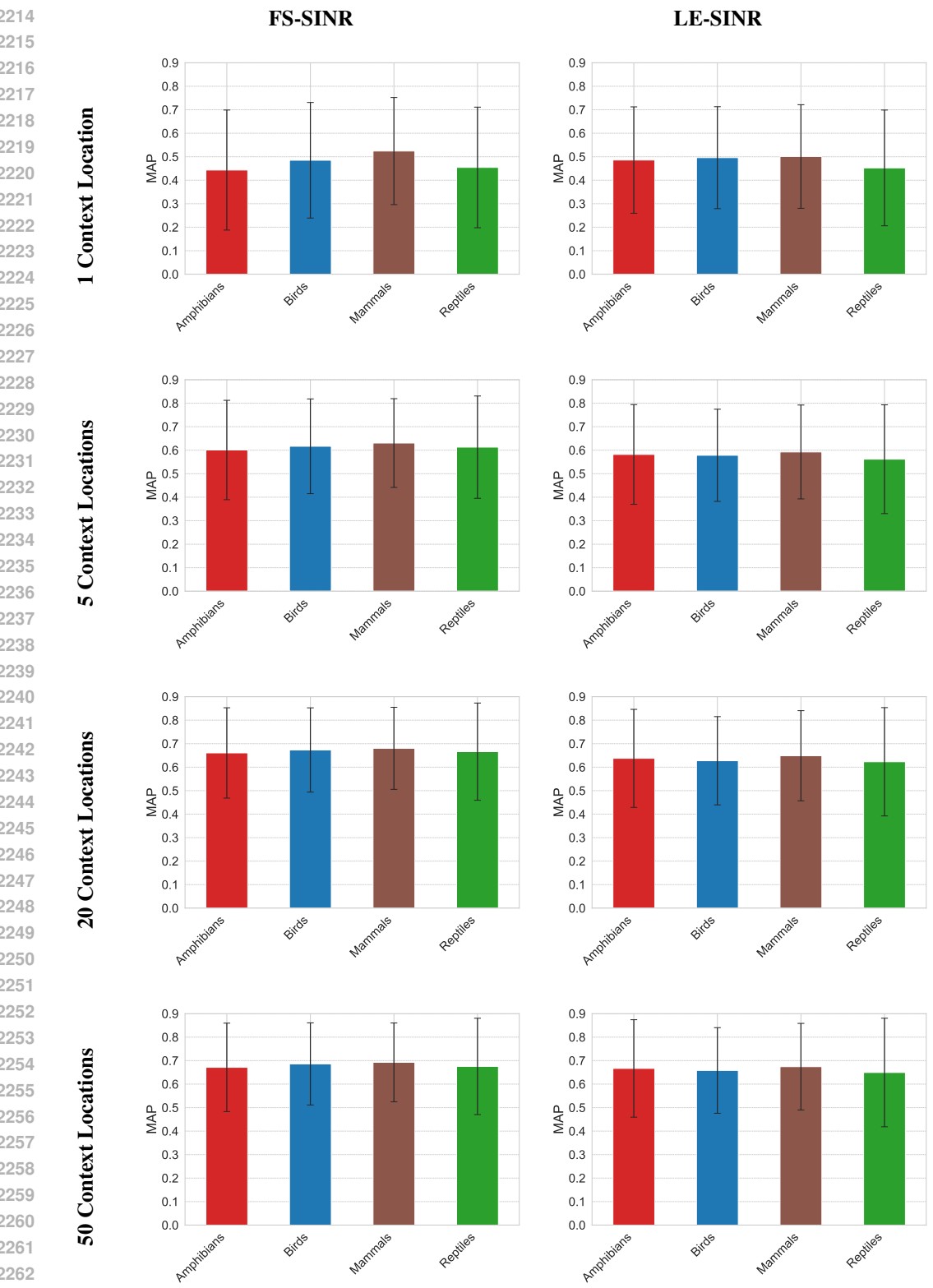

Figure A26: **Few-shot performance by taxonomic group.** Here we see few-shot IUCN evaluation results using habitat text for FS-SINR and LE-SINR for a range of context locations. Increasing the number of context locations generally increases performance across both models. We see that using small amounts of location data reduces the imbalance across taxonomic groups seen in Fig. A25, though mammals still outperform other groups slightly.

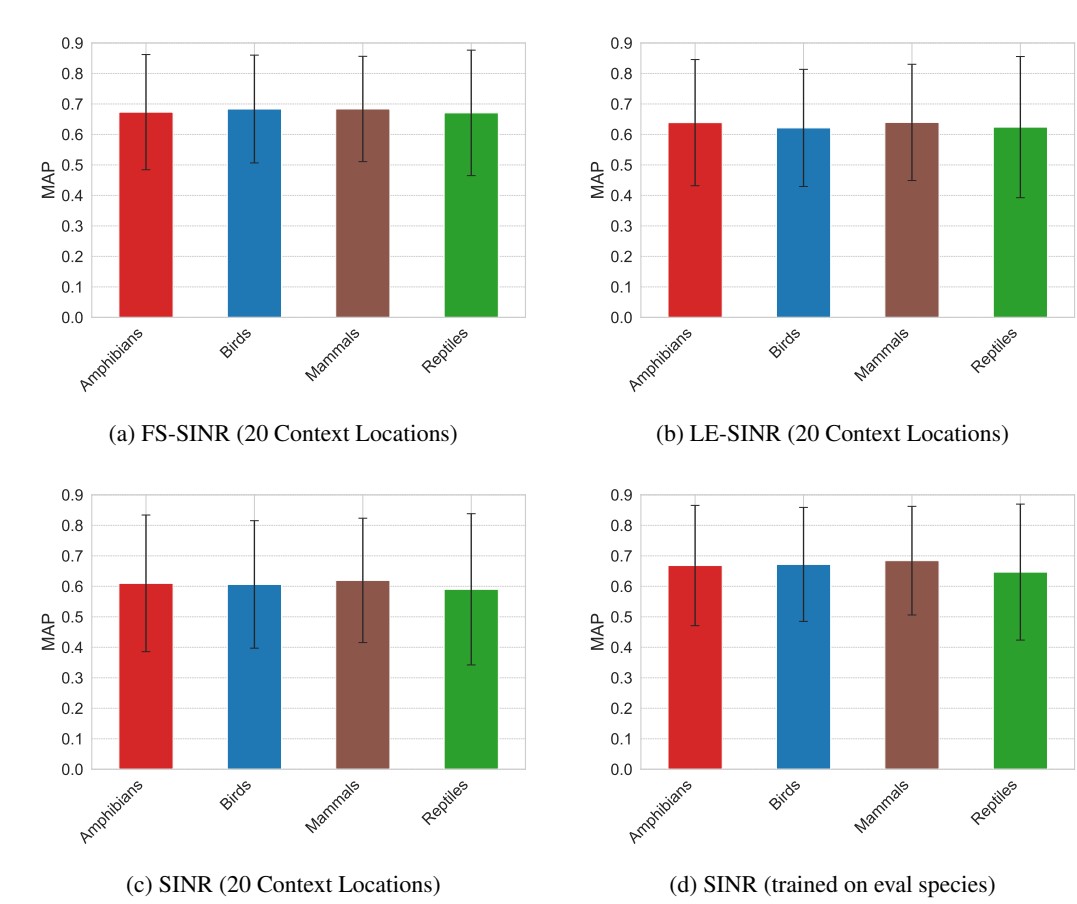

(a) FS-SINR (20 Context Locations)      (b) LE-SINR (20 Context Locations)

(c) SINR (20 Context Locations)      (d) SINR (trained on eval species)

Figure A27: **Few-shot performance by taxonomic group without text.** Here we see IUCN evaluation performance for a range of models where text was not provided during evaluation. "SINR (trained on eval species)" was trained on up to 1000 examples per-species for both the train and eval species, and the species vectors learned during training are used during evaluation, as in Cole et al. (2023). Despite this, FS-SINR still performs marginally better.

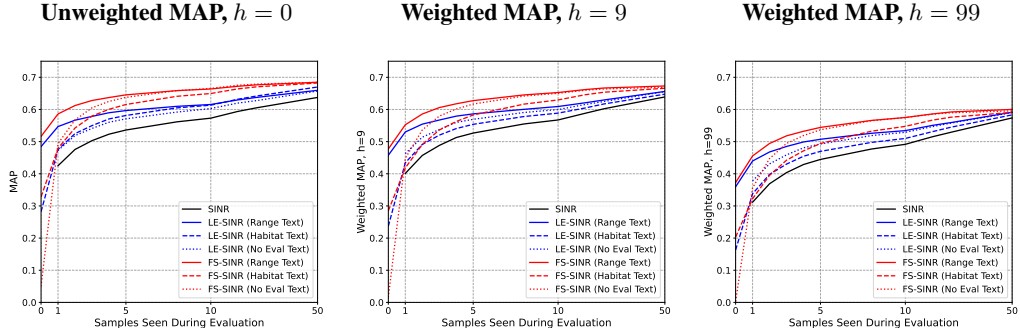

Figure A28: **Zero-shot and few-shot performance using our distance weighted MAP metric on the IUCN evaluation dataset.** We find that increasing the distance weight hyperparameter, $h$, reduces performance across the board without significantly changing the order of different models i.e., FS-SINR continues to outperform LE-SINR and SINR. We do see that approaches using habitat text decrease in performance more as $h$ increases, relative to approaches not using text or using range text.

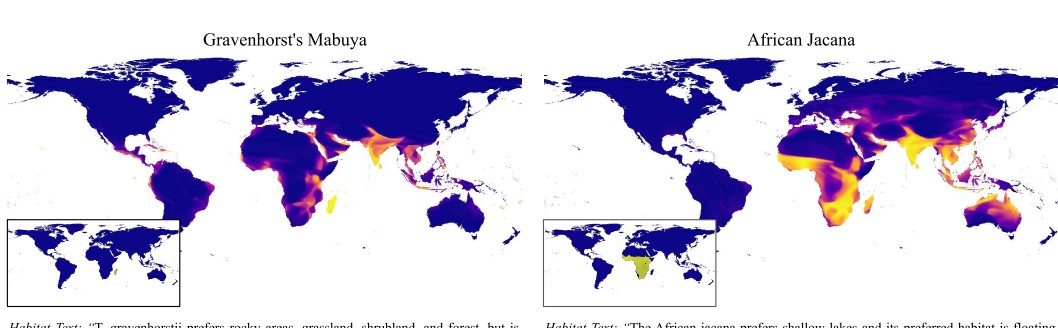

*Habitat Text:* "T. gravenhorstii prefers rocky areas, grassland, shrubland, and forest, but is also found in coffee plantations and ylang-ylang plantations. The species is adapted to a variety of environmental conditions, including different types of ecosystems."

*Habitat Text:* "The African jacana prefers shallow lakes and its preferred habitat is floating vegetation. It has a specific environmental condition, which is not mentioned in the article, but it can be inferred that it requires a certain level of water depth and vegetation cover."

Figure A29: **Examples of two species with poor distanced weighted MAP performance.** Here we visualize FS-SINR's zero-shot predictions using habitat text for two species where there is a large difference between the evaluation scores using the standard MAP metric compared to the distance weighted one (here using $h = 9$). For the `Gravenhorst's Mabuya` (left), which is endemic to Madagascar, we obtain an MAP of 0.419 but a lower distance weighted MAP of 0.175. For the `African Jacana` (right), found in most of sub-Saharan Africa, we obtain an MAP of 0.457 and a distance weighted MAP of 0.226. The distance weighted metric more heavily penalizes mistakes for these species that are very far from their true range.

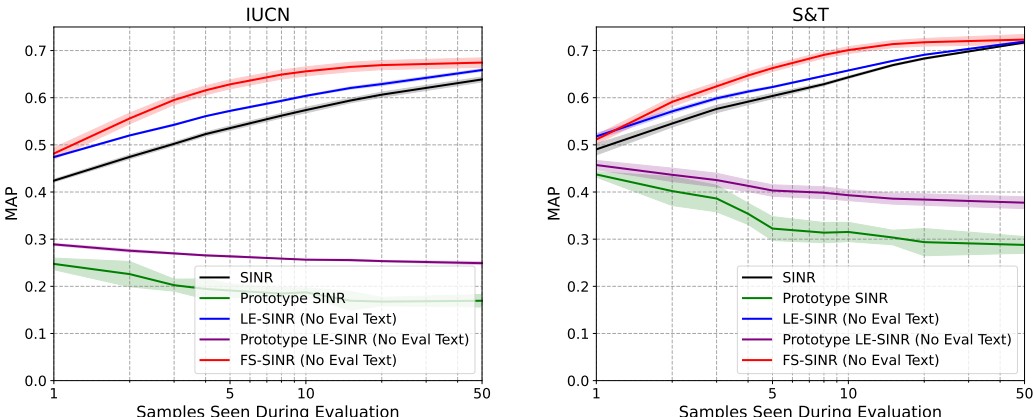

Figure A30: **Additional few-shot results using a post-hoc prototypical network type approach.** We provide two additional baselines where we freeze the backbone of the SINR or LE-SINR models and use their features to construct a prototypical type network, denoted as 'Prototype SINR' and 'Prototype LE-SINR' respectively. These two additional baselines do not require any training on the evaluation species.

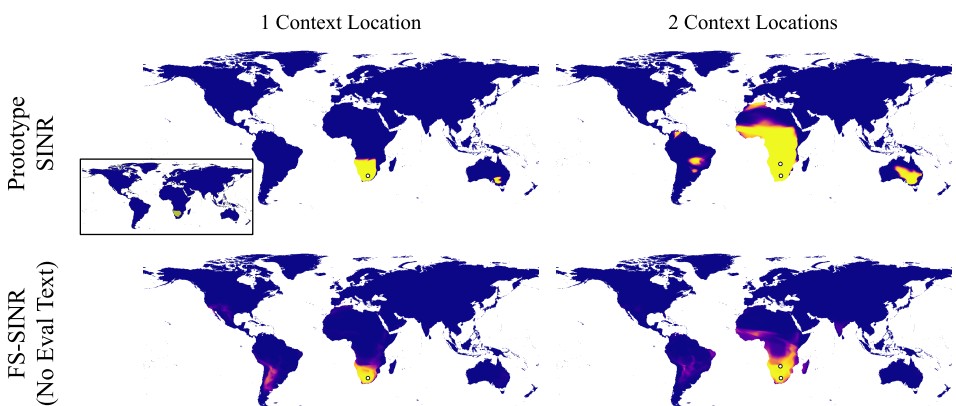

Figure A31: **Qualitative comparison of the Prototype SINR post-hoc method.** Here we compare the predictions of our FS-SINR approach (without any text) and the 'Prototype SINR' baseline on the `Kalahari Scrub-Robin` species that is found in Southern Africa. The Prototype SINR approach obtains an MAP 0.59 and 0.24, for one and two context locations respectively, i.e., it performs worse as more data is added. In the case of one context location it over predicts the range of the species and incorrectly predicts that it is present in Australia. In contrast, FS-SINR obtains 0.79 and 0.85, for one and two context locations respectively. Note, as in the other results figures, we apply an ocean mask for the visualization which hides some of the prediction errors.

