# OpenReview forum: "Few-shot Species Range Estimation"
_ICLR.cc/2025/Conference — Submitted to ICLR 2025_

### Official Review · Reviewer_uR9u · 2024-10-24

**Soundness:** 3
**Presentation:** 3
**Contribution:** 3
**Rating:** 6
**Confidence:** 4

**Summary:**

The paper proposes an architecture and a training procedure for few-shot species range estimation. The authors argue that current species observation datasets contain limited observations for majority of the species. They propose **FE-SINR**, a transformer-based architecture that can accept a few context locations and optional textual description of species and estimate ranges of unseen species. By providing a few positive locations as context, FE-SINR can estimate the range of a species more effectively than existing state-of-the-art SDMs. FE-SINR is trained on 35.5M observations in iNaturalist and evaluated against SINR and LE-SINR in both few-shot and zero-shot setting. The results reported in the paper confirm the effectiveness of the proposed method.

**Strengths:**

- The motivation and problem formulation is sound and interesting. There is a growing need for reliable species distribution models for rare and unseen species and the paper addresses the problem of unseen species distribution modeling in low data regime.
- The authors propose a transformer-based architecture that processes a few presence locations of unseen species, along with metadata and predicts the range of a given species.
- The experiments are framed and conducted well.

**Weaknesses:**

- I believe there are a few flaws in the methodology that need to be clarified:
    1. The context location embeddings provided as input to the transformer seems to be an important piece. During training, the authors use **fixed** 20 context locations as input per example. Why do the authors use 20 context locations for every species? Mallard, for example, is found in four continents, which might need larger number of context locations to cover its range as compared to *platypus*, which is only found in Australia and Tasmania.
    2. The context location encoder used in FS-SINR is a pretrained SINR (**Line 252**), which is already trained in a fully-supervised setting. Is it fair to use this location encoder in few-shot prediction scenario?
    3. Regarding inference, how do you select the optimal number of context locations to guarentee the range of a given species will be accurately mapped? Since, I see **false activations** increasing with increasing number of context locations in **Figure 4**. For example, *Black and White Warbler* getting activated in India in the bottom-most figure.
    4. Who is providing the context locations during inference? By the end user or are they retrieved?

> **Line 188-191** For the SINR model to make predictions for a new species, it is necessary to learn a new embedding vector $w_s$ for that species. If additional location data is later observed for that species, the model must be updated again.

- This is not entirely correct. There are methods like nearest-neighbor (Wang et al. (2019)) or prototypical networks (Snell et al. (2017)), which can be used as a post-hoc method for few-shot classification using frozen models. How does SINR and LE-SINR compare to FS-SINR using such methods without training?

- The performance improvements in zero-shot setting as shown in **Table 1** seems marginal. It is surprising FS-SINR performs worse than SINR even when test species are present during training (row 2 and 3). Section 4.3 on zero-shot evaluation needs more discussion and critical evaluation of the results.

- Have the authors considered comparing to the approach described in Lange et al. (2023) with a few pseudo absence locations? Since, **Line 276-280** mentions, SINR and LE-SINR are fine-tuned on few-shot presence observations with pseudo-absences.

- Some visualizations of what the latent embedding looks like after passing through the species decoder might help. How are the learned species embeddings different than the locations embeddings?, since the species ranges are predicted using the inner product between the two.

- **Section 3.2** has confusing notations. Are the networks $h_\phi$ and $m_\psi$ referring to the same thing? Please refine **Figure 2** and clearly label each component according to the notations used in **Section 3.2**.

References

Wang, Yan, et al. "Simpleshot: Revisiting nearest-neighbor classification for few-shot learning." arXiv preprint arXiv:1911.04623 (2019).

Snell, Jake, Kevin Swersky, and Richard Zemel. "Prototypical networks for few-shot learning." Advances in neural information processing systems 30 (2017).

Lange, Christian, et al. "Active learning-based species range estimation." Advances in Neural Information Processing Systems 36 (2024).

**Questions:**

How would the model incorporate absence data as context (if made available)?

---

> ### Author Response · Authors · 2024-11-23
> **uR9u rebuttal response part 1**
>
> We thank **uR9u** for their review, and respond to individual comments below.
>
> **[uR9u-1] Why 20 context locations?**
> In the new ablations results in Fig A1 in the revised PDF we observe that while there are slight improvements when using 50 context locations during training, there are diminishing returns compared to 20. Using 20 context locations is much faster to train and thus we use it for our main experiments. In general, we observe that after 20 context locations provided as input during inference we only see smaller improvements in the predicted ranges (see Fig 3). Regarding large range species, results broken down by range size can be found in Figs A22 to A24. There we observe a small drop in performance for very large range species, but this can partially be explained by biases in the training data as opposed to having not enough context locations (see Fig A20).
>
> **[uR9u-2] Use of pre-trained SINR model.**
> Importantly, the SINR encoder that we use (L252) has not observed any observations from the held-out evaluation species. As a result, it only has access to the same observation data that is available during training, i.e. this is also the same data that the baselines get. We have updated the text in Sec 4.1 to make this clear. In Fig A5 we investigate the impact of using different amounts of data to pre-train the SINR encoder, where we observe that more data improves performance as the encoder has learned a better spatial representation.
>
> **[uR9u-3] Selecting context locations at inference time.**
> For our results in Fig 3, the context locations are selected randomly from real observation data from those species from iNaturalist. We have provided a new detailed description of how this is performed in Appendix A. Unlike recent work such as Lange et al. NeurIPS 2023 who explore species range estimation in the context of active learning (but where they assume they have access to presence *and* absence data), we do not target the location selection problem. Instead, FS-SINR accepts any set in context locations as input and can make a prediction from them. Fig 3 shows that when averaged across the evaluation species, more input context locations results in better range predictions. Regarding the results for “Black and White Warbler” in Fig 4, adding more context locations is giving the model the signal that this is not just a species that has a very narrow range, but instead could be something like a bird that migrates between North and South America. There is a slight increase in the probability over India, but the scores here are much lower than those found in the Americas. Our training data contains individual species that are found on multiple continents, so predictions like this in this low data regime are not necessarily “wrong”. We also provide new qualitative comparisons in Figs A17 to A19 in the Appendix where we observe improvements as more context locations are provided.
>
> **[uR9u-4] Who is providing context locations at inference time.**
> As noted in the previous response, for the experiments in the paper, the context locations are randomly selected based on the real iNaturalist data (see Appendix A). Similarly, they are selected at random during training. However, at inference time these points could be selected using any approach (e.g. random, user selected, or more sophisticated selection). FS-SINR enables real-time interaction with users and we believe could serve as a valuable exploration tool for practitioners.
>
> **[uR9u-5] Post hoc few-shot learning using frozen models.**
> Thanks for the suggestion, we will provide comparisons to such post-hoc methods in the final version of the paper. We would expect these results to be worse than the LE-SINR and SINR models which are trained during evaluation. Note, as described in Appendix A3, when training  LE-SINR we regularize the learned weight vector so that it stays close to the initial zero-shot prediction which could be thought of as a kind of `class prototype’.

---

> > ### Author Response · Authors · 2024-11-23
> > **uR9u rebuttal response part 2**
> >
> > **[uR9u-6] Few-shot performance in Table 1.**
> > Respectfully we disagree that it is surprising that “FS-SINR performs worse than SINR even when test species are present during training”. This demonstrates that FS-SINR is not overfitting the data. Note, that in Table 1, the SINR model (1st row) has been trained end-to-end on data from the species in the test set, and thus represents a performance upper bound. In contrast, FS-SINR (rows 7 and 9) is not, and instead only gets access to a short piece of text that describes the range or habitat preferences for a previously unseen species, or in the context of rows 2 and 3, a species observed during training. If FS-SINR performed very similarly to SINR, this would suggest that it potentially is overfit to the evaluation species. We see that FS-SINR outperforms the recent LE-SINR (Hamilton et al. NeurIPS 2024). Based on your suggestion, we have updated the text in Sec 4.3 to add more discussion of the zero-shot results. We also provide an additional comparison to the type of taxonomic rank text used in LD-SDM (Sastry et al. arxiv 2023).
> >
> > **[uR9u-7] Comparison to Lange et al. NeurIPS 2023.**
> > This is an interesting suggestion. Note that the method described in Lange et al. NeurIPS 2023 requires presence *and* absence data when constructing a weighted combination of the training species, whereas the methods evaluated in our paper only make use of presence data. Additionally, their method is slow, as for each observation, one needs to evaluate all species from the training set (i.e. tens of thousands) to determine their suitability. As suggested, we will add this comparison to the final text.
> >
> > **[uR9u-8] Visualizations of the learned location encoding.**
> > We added a visualization of the learned location embedding in Fig A16 of the revised paper. There we compare to SINR, LE-SINR, and a variant of our approach that instead uses taxonomic text as in LD-SDM. We observe that the representations from LD-SDM are smooth compared to the other methods.
> >
> > **[uR9u-9] Notation in Sec 3.2 and Fig 2.**
> > $h_{\phi}()$ and $m_{\psi}()$ are very different. As noted on L139 when describing SINR, $h_{\phi}()$ is a multi-label classifier that takes a location encoding from $f_{\theta}()$ to predict the presence of S different species at a location. As noted on L203 when describing our FS-SINR model, $m_{\psi}()$ is our transformer-based encoder. Thus, $h_{\phi}()$ is not part of our model and thus not part of Fig 2, which instead uses a transformer $m_{\psi}()$. We have updated the caption for Fig 2 to make this clearer and welcome any additional suggestions for how we can make this ever clearer.
> >
> > **[uR9u-10] Incorporating absence data.**
> > This is a great question. As noted on L424 “our model is trained using presence-only data, but could be adapted to use absence information, if available, which could be denoted via a different embedding type vector.” The challenge is not incorporating the data into the model, it is instead obtaining presence-absence training data at large scale as it is very costly and time consuming to acquire. It could be possible to use pseudo-absence data instead, but this runs the risk of very strongly biasing the model. There is also potential for “semi-supervised” approaches that could use a limited amount of real absence data during training. In general, this is an exciting direction for future work that builds on our model as a starting point.

---

> ### Comment · Reviewer_uR9u · 2024-11-25
>
> I thank the authors for clarifying and updating the paper with more discussion and visualizations as requested. I have updated my score accordingly. The only remaining concern is the performance of FS-SINR in the fully supervised setting (when test species are also in the training set). I believe that overfitting in this task on training species is not necessarily a bad behavior. We are essentially trying to learn a neural field over the globe, which is based on the idea of memorization. Imagine a scenario in the future where every species on the Earth has abundant observations. In such a scenario, FS-SINR could never match the performance of SINR. I really like the idea of **FS-SINR being FS + SINR**, which is having the benefits of SINR while also adding the benefits of few-shot mapping. This idea I don't get to see from the paper.

---

> ### Author Response · Authors · 2024-11-28
>
> **[uR9u-5 - follow up] Post hoc few-shot learning using frozen models.**
> As promised, in the new Appendix D5 (see latest revision) we present new comparisons to a new Prototypical Networks-style few-shot baseline (Snell et al. NeurIPS 2017). For the new results on IUCN in Fig A30, we use a frozen SINR or LE-SINR backbone and adapt it to the few-shot setting such that it does not require any additional training on the held-out evaluation species. The model performs much worse than our FS-SINR approach. The qualitative results in Fig A31 illustrate one such failure case whereby the Prototypical Networks' predictions get worse as the features from more locations are averaged. We would like to thank In **uR9u** for this suggestion as we think it is a great addition to the paper.
>
>
> **[uR9u-11] Performance of FS-SINR in the "fully supervised setting".**
> One simple way to further improve the performance of FS-SINR would be to have a learned species embedding like in LE-SINR (see Fig 2 in Hamilton et al. NeurIPS 2024). However, this comes at the cost of significantly more parameters (as noted in Sec 4.1, FS-SINR has half the number of learnable parameters as SINR). We should also emphasize that it is only in the *zero-shot* setting, i.e. not the few-shot setting as in Fig 3, where FS-SINR performs `worse' than the SINR variant that has trained on the evaluation species (see Table 1 - row 1 vs row 3). This is because of the relatively weak signal provided by the input text. However, we can see that our FS-SINR approach, not trained on the evaluation species, obtains relatively similar performance (Fig 3 / Table A3/A4; 0.67 IUCN and 0.71 S&T) to the SINR model that *has* been trained on the evaluation species (Table 1; 0.67 IUCN and 0.77 S&T for SINR) with only 20 samples and no text.

---

### Official Review · Reviewer_NdXm · 2024-11-03

**Soundness:** 3
**Presentation:** 3
**Contribution:** 3
**Rating:** 5
**Confidence:** 5

**Summary:**

This paper addresses the task of species range estimation, which is represented as a location-conditioned probability of observing an individual of a given species. Unlike previous approaches, this work does not explicitly model a given species (such as from a species label or an image of the species). Instead, the idea is to predict a species vector from a set of sparsely sampled observations and a textual description (which could include a range or a habitat description). A location embedding is estimated based on each location of interest. The final per-location probability is computed by passing the dot product of the species encoding and the location encoding.

**Strengths:**

1. This seems like a new setting for species range estimation, addressing the important long-tail problem.
1. The writing is clear and the methods are sensible.
1. It should be easy to replicate the architecture at a high level but important details about the specific modules are missing.
1. There is some nice discussion of the limitations of the work.

**Weaknesses:**

1. The methods are sensible, but there aren't any significant theoretical/technical contributions.

1. The paper ignores some of the most important issues related to making this a practical system for real-world use cases (dataset bias and uncertainty modeling). On the uncertainty modeling side, consider Figure 4, where the difference between range maps between 1 and 5 samples is relatively minor and each has fairly confident predictions. This could be misleading for a practitioner.

1. There has been quite a bit of work on few-shot learning (see https://arxiv.org/abs/2203.04291 for a survey), but I don't see much if any of that work cited. I was expecting to see some connections to that literature on general approaches to few-shot learning to place this approach in context. Also, some indication of why this particular approach was best for this task.

1. I would have liked to see some qualitative comparisons between maps constructed by the baselines and the proposed methods. That would help better understand the reason for the improved performance.

**Questions:**

1. Is MAP a meaningful error metric for this task given that it ignores geographic distance?

1. I don't fully understand how the few-shot experiments for SINR were conducted. Was the model retrained including the full dataset plus the restricted (<k) samples for the evaluation species? Or, was the model limited to at most k samples for all species and trained at that level? I see that there are experiments with random initialization and without, so perhaps the results in the main paper are from a pre-trained SINR with then separate fine-tuning for each species? Or, fine-tuning across all evaluation species?

1. What is the practical impact of the register token? Was this evaluated?

1. Will sufficient technical details be released to exactly replicate this work, either source code or architecture and training details? What exactly is the transformer block? The project head? How is training performed (optimizers, etc.)?

1. Figure 5 highlights a concern I have about several of the results. It seems like the model does not put much weight on individual observations. For example, the "desert" case shows that the observation was in a very low probability region.  Also, it would be nice to see what these figures look like across the globe. Some of the results in the appendix (A.8) indicate that the model doesn't seem well suited to modeling endemic species. I suspect if you zoomed out from the Figure 5 examples you would see that it was mostly highlighting areas that match the range description.

---

> ### Author Response · Authors · 2024-11-23
> **NdXm rebuttal response part 1**
>
> We thank **NdXm** for their feedback, and respond to individual comments below.
>
> **[NdXm-1] No significant theoretical/technical contributions.**
> We present a new approach for few-shot species range estimation and we show that it outperforms existing state-of-the-art methods (see Fig 3). Our main technical contribution is a novel architecture for this task that enables few-shot estimation in a single forward pass (Fig 2) without requiring any retraining. Existing methods such as SINR (Cole et al. ICML 2023) and LE-SINR (Hamilton et al. NeurIPS 2024) can only be applied to the few-shot setting with additional retraining. Species range estimation is an important problem, and we believe that the few-shot setting will significantly benefit from input from the machine learning community.
>
> **[NdXm-2] Dataset biases.**
> As noted on L426, like several previous neural-network methods (e.g. Cole et al. ICML 2023, Lange et al. NeurIPS 2023, Sastry et al. arxiv 2023. Hamilton et al. NeurIPS 2024, …) we do not explicitly account for spatial biases in the data. There have been some attempts in the machine learning literature to address this (e.g. Chen et al. AAAI 2019). We agree that this is indeed an important consideration, we believe that it is a complementary research direction to the one explored in our work which is focused on few-shot estimation. Future advances in loss functions and de-biasing techniques could be combined with our model.
>
> **[NdXm-3] Uncertainty modeling and the difference between 1 and 5 context locations.**
> In the results in Fig 3 we observe large differences in MAP when comparing models that have 1 versus 5 context locations. For example, for “FS-SINR (No Eval Text)”, there is a difference of ~0.15 MAP on the IUCN data. While they may appear small at a global scale, in Fig 4 we do see noticeable differences between the model predictions when the number of context locations is increased. Fig 5 also demonstrates that large differences in the output predictions can be observed when changing the input text. This demonstrates that our model is responsive to user guidance. Additional new results in Figs A17 to A19 also show how the predicted ranges become more refined as more context locations are provided. However, we also agree with the suggestion that it would be beneficial to have stochasticity in the model predictions. As noted on L420, this could be achieved via an additional sampling layer. To demonstrate that FS-SINR does produce varying outputs, we present new results in Fig A15 where we show that different random initializations (i.e. different random seeds) during training result in very different predictions in the zero-shot setting.
>
>
> **[NdXm-4] Related work on few-shot learning.**
> L108-L123 discussed related work on few-shot learning in the context of species range estimation. There are many aspects of the problem that make it distinct from few-shot learning explored in other domains (e.g. in image classification). For one, the input domain is fixed (i.e. all the locations on earth), each location can support more than one species (i.e. multi-label as opposed to multi-class), the label space is much larger (i.e. tens of thousands of species as opposed to hundreds of classes in image classification), and only partial supervision is available (i.e. we only have presence data, with no confirmed absences). We agree that there could be many exciting connections to the larger literature on few-shot learning and hope that our work, and benchmarking, motivates further research in this direction. To address your comment, we have updated the related work section to make these distinctions clearer. Thanks for the suggestion!
>
> **[NdXm-5] Additional qualitative results.**
> We have provided a new section of additional qualitative and quantitative results in Appendices C and D respectively. There, we demonstrate additional qualitative comparisons to LE-SINR and SINR (e.g. see Figs A17 to A19) and LD-SDM-like taxonomic text in Fig A9.
>
> **[NdXm-6 Q1] MAP ignores geographic distance?**
> This is a good suggestion. For fair comparison to existing work, we use MAP as it is the standard metric used by the recent baselines we compare to (e.g. SINR, LE-SINR).  LD-SDM (Sastry et al. arxiv 2023) proposes an alternative metric which takes the distance from the nearest observations into account. In their work, it is applied in the context of presence-only evaluation data. There are some design decisions that are important to take into account if one was to adopt their metric. For example, should one measure distance from a test location to the closet, or average, presence location. We will implement a weighted MAP metric and report results in the final version of the paper as it is not possible now due to time constraints. Based on the qualitative results already in the paper, we do not expect that this metric will result in any differences between the rankings of methods.

---

> > ### Author Response · Authors · 2024-11-23
> > **NdXm rebuttal response part 2**
> >
> > **[NdXm-6 Q2] How were the few-shot examples for SINR constructed?**
> > For the results for SINR in Fig 3, we follow the same procedure as LE-SINR (Hamilton et al. NeurIPS 2024) and freeze the backbone network and train a linear classifier (i.e. a logistic regressor) with background pseudo-absences for each few-shot evaluation species (as noted on L275). The SINR model is pre-trained on all the species in the train set (i.e. it does not include those in the evaluation set) (L269). We have updated the implementation details text in Appendix A to make it clearer how training and evaluation locations are sampled.
> >
> > **[NdXm-6 Q3] Impact of register token.**
> > As requested, in Fig A6 we ablate various model choices for FS-SINR (results are averaged over three runs). While the register token only provides a relatively minor benefit, we observe that not using the token type embeddings results in the model not being able to learn.
> >
> > **[NdXm-6 Q4 - part 1] Additional implementation details.**
> > We have updated Appendix A to more clearly describe the implementation details. There we provide information about model architecture, training, and evaluation.
> >
> > **[NdXm-7 Q4 - part 2] Code release.**
> > Yes, our plan is to release the code and pre-trained models to ensure reproducibility. We strongly believe that this is an important problem that needs input from the machine learning community. Our hope is that our model and benchmarking serves as a future baseline for others to make progress on this task.
> >
> > **[NdXm-8 Q5 - part 1] Impact of individual observations.**
> > As noted in response [NdXm-3], in the low data setting we observe big increases in model performance as the number of context locations is increased. The example in Fig A11 demonstrates that adding context locations (e.g. first in North Africa, then in South America, and finally Australia) significantly alters FS-SINR’s predictions. In other cases where the initial prediction (e.g. from the text input) is reasonable, then large changes in the predictions are not so apparent. An example of this is Fig A17, where the model broadly gets South America correct but is producing false positive predictions into Central America (see left column, top row). As more context locations are provided, these errors are refined (see left column, bottom row). Again, these observations are consistent with the results in Fig 3 where we see big changes for fewer numbers of observations and a reduced benefit to adding more (i.e. > 20). As requested, we will also provide the global predictions for Fig 5.
> >
> > **[NdXm-8 Q5 - part 2] Results on endemic (i.e. small range) species.**
> > To address this question, in Figs A22 to A24 we provide a detailed evaluation of model performance binned according to the range size of species in the evaluation set. In Fig A23 we observe that FS-SINR results in superior performance compared to LE-SINR for almost all range sizes, even small ones (i.e. < 10k and < 100k km^2).

---

> ### Author Response · Authors · 2024-11-28
> **NdXm rebuttal response part 3**
>
> **[NdXm-4 - follow up]  Related work on few-shot learning.**
> In Appendix D5 we present new comparisons to a new Prototypical Networks-style few-shot baseline (Snell et al. NeurIPS 2017). This was suggested by **uR9u**. For the new results on IUCN in Fig A30, we use a frozen SINR backbone and adapt it to the few-shot setting such that it does not require any additional training on the held-out evaluation species. The model performs much worse than our FS-SINR approach. The qualitative results in Fig A31 illustrate one such failure case whereby the Prototypical Networks' predictions get worse as the features from more locations are averaged.
>
> **[NdXm-6 Q1 - follow up] MAP ignores geographic distance.**
> As promised, we implement the suggested distance weighted evaluation metric. This was something that **vJfa** also wanted to see based on **NdXm**'s review. We think this is a great addition to the paper and we thank **NdXm** for suggesting it. As predicted the new results on the IUCN dataset in Fig A28 demonstrate that the rankings of different methods are not impacted by the weighting. However, overall performance does decrease for all methods.  We describe the new distance weighted metric in Appendix D4 and illustrate qualitative examples in Fig A29.

---

> > ### Author Response · Authors · 2024-12-01
> >
> > Greetings **NdXm**,
> >
> > The discussion period is coming to a close soon. Please do not hesitate to follow up if you have any additional comments regarding the updates we have made to the paper based on your requests. Thanks!

---

### Official Review · Reviewer_vJfa · 2024-11-03

**Soundness:** 3
**Presentation:** 4
**Contribution:** 3
**Rating:** 6
**Confidence:** 4

**Summary:**

This paper presents a method for few-shot species range estimation building on the SINR paper and the idea of adding context information in the form of textual information about range or habitat as was presented in the LE-SINR paper. Tokenized locations where a species has been observed are combined with context information in a transformer module along with a class token and register token. For a new location, the output class token of the transformer is passed though a small MLP and combined with the embedding of the query location to produce the final prediction.

**Strengths:**

- The paper is overall clear and well-written
- The main limitations are well identified.
- Several ablation studies are provided to anticipate different questions that could arise.
- The problem that the paper tackles (species range estimation in the few-shot setting) has not been explored much before but could have important impact in ecology.
- The proposed method outperforms other existing methods (which haven't been designed for the context of FS learning specifically) but more importantly enables predicting range maps for previously unseen species without any retraining, which could make it appealing in a real-life application setting.

**Weaknesses:**

- Limited technical novelty and limited ecological analysis: this paper builds on LE-SINR and SINR, the main difference being the stack of transformer blocks used to combine text and location data. I also understand this work builds on previous work, and thus it is common in the machine learning context to not question certain design choices (the choice of LLM for the text encoder, the evaluation metrics, the different types of analyses done), but as this is a different application context that is put forward (few-shot setting), it would make the paper really more convincing (and appealing to ecologists who might use this method) to dive deeper in an ecological analysis of the results. (see some questions in the questions section)

- One of the main limitations I see of this work is pointed out by the authors themselves, i.e. the fact that the same set of locations will always give the same range map. Proposing different possible range maps could definitely be more appealing to downstream users of such a method, especially when very few locations are provided.

**Questions:**

- I would like to confirm the methodology used for doing the different runs that make up the error bars in the few-shot setting.  For a given test species,  are the provided context locations of the different runs different?

-   Have the authors looked into how /if  the performance of the model changes depending on the type of habitat /  the more or less restricted range that species have?

- It seems in the Black and White Warbler example that predictions in South America go up as more examples in North America and Central America are added. As the authors point out, the method could be more effective in practice if a way to handle absences is added.
I wonder if there is a way to consider adding information not only about range and habitat but also about the family/ genus of the species, and whether that could help having more precise predictions.

- The following question is connected to the previous point. I understandable this work builds upon previous SRM estimation papers using their datasets and metrics, but I wonder if other evaluation metrics than MAP in the few-shot setting could be added.
Have the authors looked into analyzing the performance by taxonomic group / geographical distribution of the species? I suppose there is some bias in the data used for training (the majority of the data would be for species observed in North America and Europe) and it would be valuable to understand to what extent this method works for less well surveyed regions  because from a practical standpoint it seems those regions would be where few-shot learning would be the most helpful.
To phrase it differently, this paper frames an important and relevant problem but if such a method is to be truly used in a real-life context, some more focus could be put into showcasing examples that are ecologically relevant, for example showing the performance for different groups of species depending on the number of observations available in the training set.

- In general, it would be interesting to have a bit more ecologically relevant analysis of the results. I really appreciate the effort of the authors to show examples of concepts learned by the text encoder in the appendix, but it would perhaps resonate more in the ecology community if there was some analysis showing to what extent the language encoder learns taxonomical hierarchy.



Minor comments about figures:
- Unless the locations overlap, it seems the same figure is provided for Hyacinth Macaw on rows-5 of Figure 4. Perhaps it would be helpful to just zoom into the South American region (I don’t see 5 or 10 context locations)
- The FS-SINR (No Train or Eval Text) model line is quite hard to see in figure A1
- Numeric results of Figure 3 could be reported in a table in the appendix to make it easier to read the overlapping error bars of different models as the number of samples increases.
- In the appendix figures A7 and A8 it would be nice to show that the textual descriptions are for range and habitat.
- I would have liked to see some error bars for the ablation studies. I understand this is appendix content, but it makes it a bit difficult to agree fully with certain statements, e.g. in figure A2: "we observe the trend that as more data is  used, performance increases" It seems that allowing for a higher maximum number of  training examples during training does not necessarily lead to better performance in the case when  textual information is provided in the form of range or habitat description. I read the figure as showing that initializing from different SINR models that have seen greater number of examples for each species does not make much of a difference when textual data is provided.

---

> ### Author Response · Authors · 2024-11-23
> **vJfa rebuttal response part 1**
>
> We thank **vJfa** for their detailed comments and feedback. We respond to individual comments below.
>
> **[vJfa-1] Limited technical contribution.**
> We present a new approach for few-shot species range estimation and we show that it outperforms existing state-of-the-art methods (see Fig 3). Our main technical contribution is a novel architecture for this task that enables few-shot estimation in a single forward pass (Fig 2) without requiring any retraining. Existing methods such as SINR (Cole et al. ICML 2023) and LE-SINR (Hamilton et al. NeurIPS 2024) can only be applied to the few-shot setting with additional retraining. Species range estimation is an important problem, and we believe that the few-shot setting will significantly benefit from input from the machine learning community.
>
> **[vJfa-2] Provide more ecological analysis.**
> Thanks for this suggestion! Based on this request, we have added much more analysis to the Appendix to provide a deeper understanding of the performance of our method. In Fig A16 we visualize the learned spatial representation which demonstrates that geographic features emerge during training like in SINR. In Fig A20 we visualize false positives on the map where we see more error in Africa and South America compared to Europe and the US. This is likely explained by spatial biases in the data. In Fig A21 we visualize the same types of errors for models that either use range, habitat, or no text. In Figs A22 to A24 we report performance per-species binned by their range size, where we observe that small and very large range species are challenging. Finally, in Figs A25 to A27 we report performance across coarse taxonomic groups from the evaluation set (i.e. amphibians, birds, mammals, and reptiles), where we observe that there are no major differences in performance across different types of species. We hope that this additional analysis will make our work more appealing to ecologists who might use it.
>
> **[vJfa-3] Model output is deterministic.**
> The current model is deterministic, but as we see in the example in Fig A11 adding context locations (e.g. first in North Africa, then in South America, and finally Australia) significantly alters FS-SINR’s predictions. As noted on L420, this could be achieved via an additional sampling layer. To demonstrate that FS-SINR does produce varying outputs, we present new results in Fig A15 where we show that different random initializations (i.e. different random seeds) during training result in very different predictions in the zero-shot setting.
>
> **[vJfa-4] How are the error bars generated?**
> We have updated the description in the implementation details in Appendix A to clarify this point. Specifically, we perform three runs for each experiment using different seeds and report the mean. We display  the standard deviation as error bars in our figures.
>
> **[vJfa-4] How does performance depend on the type of habitat or range size?**
> Please see the response to query [vJfa-2], where we point to new results that break down performance based on geographical location (Fig A20) and range size (A22 to A24)
>
> **[vJfa-5] Incorporating coarser taxonomic information.**
> Based on this suggestion we added new comparisons to the type of taxonomic rank text used in LD-SDM (Sastry et al. arxiv 2023). Instead of using free-form text as input, this involves using a text string that encodes the taxonomic hierarchy of the species (i.e. species, genus, …). New zero-shot results are provided in Table 1 in the main paper with a full breakdown per-rank level in Table A2. More details and visualizations (e.g. Fig A9) of these experiments are provided in Appendix B7.
>
> **[vJfa-6] Has the language encoder learned taxonomic hierarchy.**
> In Fig A8 we observe that the model that uses Wikipedia-derived free-form text outperforms all forms of taxonomic text, with range text being superior. However, as the number of content locations increases the performance difference is smaller. While this does not establish if the language encoder represents taxonomic hierarchy, it shows that free-form text results in superior range predictions in the few-shot setting.

---

> ### Author Response · Authors · 2024-11-23
> **vJfa rebuttal response part 2**
>
> Additional comments about the figures:
>
> **[vJfa-7] Qualitative results for Hyacinth Macaw in Fig 4.**
> In the case of the Hyacinth Macaw (right column) in Fig 4, there is strong spatial bias in the data whereby the overwhelming majority of observations for the species come from the south west of Brazil, while the species has a larger range encompassing remote locations. This can be seen clicking on the “Map” tab on iNaturalist for this species:
> https://www.inaturalist.org/taxa/18938-Anodorhynchus-hyacinthinus
>
> As a result, random sampling of locations for this species results in very similar spatial locations being returned. See our response [uR9u-4] for more discussion of sample selection.
>
> **[vJfa-8] Figure clarity in Appendix.**
> We have significantly updated the Appendix such that the results are better organized and easier to follow.
>
> **[vJfa-9] Report the results from Fig 3 in a table in the Appendix.**
> Good idea. We have added these results to Tables A3 and A4 in the Appendix.
>
> **[vJfa-10] Add the text descriptions for Figs A7 and A8.**
> We have added the habitat and range text for these species in the figures (now A13 and A14).
>
> **[vJfa-11] Error bars for the ablations in the Appendix.**
> As requested, we reran and updated the ablations and added error bars. For clarity, we present results with and without error bars so that it is easier to see the results of different methods.

---

> > ### Comment · Reviewer_vJfa · 2024-11-26
> > **response to rebuttal**
> >
> > I thank the authors for the additional details on the experimental setup and the analyses that were provided, as well as the responses to the other reviews. All the additional visualizations are really appreciated and improve the paper, so I am updating my score.
> >
> > However, echoing reviewer NdXm, I am still not entirely convinced about the choice of mAP as a sole metric in this context, even though this was what was used in other baselines. It doesn't reflect the overestimation of range that can occur (as pointed out in the example in  A19), but more importantly in the context of FS SDM, it seems one could want to have an accurate local range map around the locations that are provided - so having some sort of geographically weighted score could be really helpful. For example, if ecologists were to use this method, they are likely to have a small very localized dataset of sample points in a specific region they are interested in, so it would make sense to make sure that the predictions are good for that region.

---

> > > ### Author Response · Authors · 2024-11-28
> > > **new results with requested geographically weighted metric**
> > >
> > > **[vJfa-12]  Geographically weighted metric.**
> > > We thank **vJfa** for emphasizing the importance of a geographically weighted evaluation metric. In Section D4 we describe a new metric to address this request. There we show new results using a distance weighted MAP metric, whereby mistakes that are further away from the evaluation presence locations, for each species, in the expert derived range maps are more heavily penalized than mistakes that are closer. This captures the intuition that it is bad to have a species incorrectly predicted in a different continent than where it is actually found (i.e. a false positive for a species in a desert in Africa, that is normally found in a desert in the US). Results are presented on the IUCN evaluation set in Fig A28. Fig A29 illustrates an example of two species where the performance difference between the standard MAP and the distance weighted variant is large. As can be seen in these examples, these are cases where there are large numbers of false positives far from the expert-derived range.
> > >
> > > Note, in our case the weight is larger the further a location is from the closest presence observation for the species of interest in the evaluation set. As suggested by **vJfa**, it would also be possible to invert the weights, such that mistakes that are closer are penalized more (i.e. more local ones). However, it is challenging to very precisely define the range of a species at a local level (e.g. a given species' range can change over time or due to factors such as climate change and habitat loss) and thus penalising mistakes close to the current evaluation data runs the risk of incorrectly penalizing false negatives.

---

### Official Review · Reviewer_EQN8 · 2024-11-09

**Soundness:** 3
**Presentation:** 3
**Contribution:** 2
**Rating:** 6
**Confidence:** 4

**Summary:**

The authors tackle the problem of predicting the geographic range of a biological species given a small set of opportunistic observations, thus focusing on the few-shot setting. This is done via in-context learning: at training and inference time, a set of presence locations are given to the model, along with a textual description of the species range or habitat. This means that it is possible to obtain predictions for a species not seen during training, as long as a few presence locations and/or textual description are available.

**Strengths:**

1) The paper is well written and structured.
2) Instead of learning a species embedding per species, as done in LE-SINR, the authors propose to use a few (20) presence locations as the species embedding. This is makes FS-SINR naturally adapted to few-shot in-context learning.

**Weaknesses:**

1) The novelty wrt LE-SINR seems to be quite limited, and is not explicitly addressed. Section 3.2 clearly addresses the contribution wrt SINR, but I could not find the same for LE-SINR. Other than the fact that FS-SINR uses a transformer to combine the different data modalities, the main difference is that FS-SINR makes use of 20 locations per species during training. However, the ablations don’t explore which of the components contribute the most to the improved results. This could be easily verified by using a learnable species token that can be given to the transformer, rather than the context locations.
2) Even though the fact of using in-context locations is the main contribution of this work, I found very little detail about this aspect. The ablation studies do show that the improvement of adding more locations slows down after 20 are added (and up to 50 are studied), but I could not understand if the authors studied using more than 50 in-context locations. In addition, I could not find information about how these locations are selected, and whether the same set is kept during the whole training procedure, or whether the in-context locations are dropped from the training set.

**Questions:**

1) Is the improved accuracy due to the in-context locations or due to the transformer model used to get the species embedding? The ablation study suggested in the previous section could help answer this question.
2) How exactly are the in-context locations dealt with during training? How much does a different choice of test-time locations affect the results?

---

> ### Author Response · Authors · 2024-11-23
> **EQN8 rebuttal response part 1**
>
> We thank **EQN8** for their detailed comments and feedback. We respond to comments individually below. One potential misconception that we would like to clarify that is related to more than one comment is that in our few-shot setting the species in the evaluation set are disjoint from those in the train set (L84 and L269 in the original submission), i.e. our evaluation species are not part of the training set during model training.
>
> **[EQN8-1] Novelty wrt LE-SINR (Hamilton et al. NeurIPS 2024).**
> The main similarity between our FS-SINR approach and the recent LE-SINR (Hamilton et al. NeurIPS 2024) is that both can use text data to assist with species range estimation. However, LE-SINR is not the first method to text, e.g. LD-SDM (Sastry et al., 2023 arXiv 2023) also uses text, albeit in a more restricted form. Otherwise, our setting is very different from LE-SINR. Specifically, we perform *few-shot* species range estimation without any retraining, whereas they are addressing the *zero-shot* setting. Their method cannot directly incorporate location observations without retraining a linear classifier. On L118 of the original submission we note the main distinction to LE-SINR where we say that “LE-SINR performed few-shot experiments whereby they used a language encoder to estimate an initial encoding for a species and combined it with a linear classifier that needs to be trained to generate range predictions.“ On L84 we note that our main contribution is that we can “predict the spatial range of a previously unseen species at inference time without requiring any retraining.” Our results in Fig 3, demonstrate a significant improvement over their method, where we observe that our predicted ranges are quantitatively superior, while also not requiring any retraining. We also see in the new qualitative examples in Appendix C2 (i.e. Figs A17, A18, and A19) especially in the low observation setting, that show that our outputs are qualitatively better. As suggested in the reviewer comment, we have updated the text in the related work section so that this distinction is much clearer. Thanks for the suggestion.
>
> **[EQN8-2] Ablations to demonstrate which of the components contribute the most to the improved results.**
> We have added a much more detailed set of ablation experiments in Appendix B to justify the main modeling components used in FS-SINR. There we explore the impact of multiple factors such as the amount of training data, input features, architecture choices, number of context locations, and different combinations of context information.
>
> **[EQN8-3] Provide a learnable species token as input to FS-SINR.**
> One of the main advantages of our FS-SINR approach is that it has significantly fewer parameters compared to SINR. In total, our model has 6.3M learnable parameters compared to 11.9M for SINR. Note, in the original submission (L255) we incorrectly stated this was 29M due to an error in how we counted the parameters in PyTorch. The challenge with the suggestion of providing a learnable species token for each of the training species is that it will then not be possible to evaluate held-out species that are not observed during training. As noted on L269, we hold out the evaluation species from the train set and thus no observations from them are observed during training. Please do not hesitate to let us know if we have misunderstood your suggestion.
>
> **[EQN8-4] Adding more than 50 context locations.**
> The most amount of context locations we evaluate with is 50. We justify this choice because we are operating in the few-shot setting. If more than 50 observations were available for a species, it would make more sense to add them to the training set. While the performance increase does plateau after a set number of observations (see Fig 3), interestingly, we observe that our approach performs very close, and sometimes even better, compared to the SINR baseline that has seen observations from the test species during training (i.e. first row in Table 1). For example, the SINR model obtains MAP of 0.67 on the challenging IUCN dataset, whereas our FS-SINR approach obtains a similar score with only 20 observations (see Fig 3 - left). Importantly, FS-SINR only requires a single forward pass through the model to obtain this score (L274). In the updated paper we provided additional results where we ablated the impact of different numbers of context locations at training time (see Appendix B.1).

---

> > ### Author Response · Authors · 2024-11-23
> > **EQN8 rebuttal response part 2**
> >
> > **[EQN8-5 - Q1] Is the improved accuracy due to the in-context locations or due to the transformer model used to get the species embedding?**
> > As noted in the EQN8-3, if we provided a learned species embedding to the transformer it would not be possible to perform few-shot evaluation for species not observed during training. However, the results in Table 1 partially address this question as we can compare the zero-shot results of a FS-SINR model with range text that has seen test species data during training (3rd row) to a version of the model that has not observed any data from those species (9th row). In this zero shot setting, we only see a relatively small drop in performance (e.g. 0.55 vs 0.52 on IUCN). This indicates that training on species does help, but is not very impactful given the weak information that is contained in the text. In contrast, the SINR model that has a learned encoding for the training and evaluation species (1st row) performs much better (0.67 on IUCN). This indicates that our transformer-based model has not overfit to these species.
> >
> > **[EQN8-6 - Q2 - part 1] How are context locations selected and are context locations dropped from the training set?**
> > At evaluation time, as in LE-SINR, context locations are randomly sampled from the iNaturalist dataset for the species in the evaluation set. We provide the same set of observations to each method we evaluate and the larger sets of context locations are supersets of the smaller ones (L279). Importantly, there is no overlap between the training and evaluation species (L269). During training, we select 20 context locations per training instance (L262) (except in the case of the ablations in the Appendix B.1 where we use more or less). We have added new text in Appendix A to clearly describe how locations are sampled during training and evaluation. Please let us know if any of these steps remain unclear.
> >
> > **[EQN8-7 - Q2 - part 2] How much does a different choice of test-time locations affect the results?**
> > Unless stated otherwise the results in the main paper and Appendix are computed by averaging over three different runs/seeds (L342). The error bars (e.g. in Fig 3) are relatively small. In the new qualitative results in Fig. A.15 we show the impact of the different random seeds used to train FS-SINR which illustrate the variation produced by different models. The new qualitative results in Figs A17 to A19 give a sense of how the model predictions change as different context points are added. Similarly, in Fig 4 in the main paper we can see when comparing the second and third row for the Black and White Warbler the impact of having a point in North Mexico versus not. These results indicate that the model is responsive to the precise choice of test-time locations selected.

---

> > > ### Comment · Reviewer_EQN8 · 2024-11-24
> > >
> > > I would like to thank the authors for the work behind their responses. I particularly appreciate the new details in the appendix. I would now rather be in favour of accepting this paper.

---

### Author Response · Authors · 2024-11-23
**Rebuttal response**

We thank the reviewers for their detailed comments and suggestions. In light of the questions related to implementation details and ablations, we have significantly updated the Appendix to include more detailed results and quantitative and qualitative comparisons to existing methods.

We have also updated the paper text based on revised comments and noted changes from the original submission using red text. Note, that in cases where the caption of a Table or Figure is all red, this indicates that the Table/Figure itself is new (with the exceptions of Fig 2 in the main paper). When referencing line numbers in our individual response to reviewers, we point to lines in the original submission.

We respond to individual reviewer comments below. Please do not hesitate to follow up with additional questions if there is anything that you would like further clarified.

---

> ### Author Response · Authors · 2024-11-28
> **Rebuttal response - follow up**
>
> We thank the reviewers for engaging in the discussion and for their valuable comments which have contributed to improving the paper greatly. Based on the suggestions we provide a final set of requested comparisons. These can be found in the revised PDF.
>
> **New evaluation metric.**
> In Appendix D4 we present new results using a distance weighted variant of the main evaluation metric we used. This was in response to a request from **vJfa** and **NdXm**. From the results in Fig A28 we observe that the relative ordering of the different methods does not change when using this new metric, and our FS-SINR approach still performs best.
>
> **New few-shot comparisons.**
> In Appendix D5 we present additional comparisons to a Prototypical Network-style post-hoc baseline. This was requested by **uR9u**.  The results in Fig A30 demonstrates that FS-SINR outperforms it.

---

### Comment · Area_Chair_xMxE · 2024-11-24
**Discussion**

Dear reviewers,

Thank you for your contribution. Soon the discussion period is about to end. Only one reviewer has responded to the author's response. We are requested other reviewers to please go over the response from the authors and initiate discussion.

regards

AC

---

### Meta-Review · Area_Chair_xMxE · 2024-12-22

**Metareview:**

Dear authors,

Thank you for submitting the draft. This draft has received slightly mixed review, with 1 reviewer assigning score 5 and rest assigning score 6 (marginally above acceptance level). Two of the reviewers that assigned 6 rating shared their concerns about the quantitative results, even when one thought "paper is an interesting contribution to species distribution modelling".

Therefore, even when the problem is quiet interesting and quite relevant to world we live in, we believe draft is not ready for acceptance to ICLR at this stage. We encourage authors to update the draft, and are hopeful that comments by reviewers will be helpful.


regards

AC

**Additional Comments On Reviewer Discussion:**

Authors shared updated results, especially the implemented they named "distance weighted evaluation metric", to answer concerns of multiple reviewers.
EQN8,uR9u and vJfa increased their rating after the author feedback. However, they only assigned rating 6 (marginally above acceptance level). vJfa shared the concerns raised by uR9u.

---

### Decision · Program_Chairs · 2025-01-22

Reject